

**PMF-LP: the first 10 m plastic-mulched farmland distribution map (2019-2021) in**
**the Loess Plateau of China generated using training sample generation and**
**classifier transfer method**
Cheng Zhao [1, 2], Yadong Luo [1, 2], Xiangyu Chen [1, 2], Linsen Wu [1, 2], Zhao Wang [3], Hao Feng [2, 4],
Qiang Yu [3, 4], Jianqiang He [1, 2, 3, *]
[1] Key Laboratory for Agricultural Soil and Water Engineering in Arid Area of Ministry of Education, Northwest A&F
University, Yangling 712100, China
[2] Institute of Water-Saving Agriculture in Arid Areas of China, Northwest A&F University, Yangling 712100, Shaanxi,
China
[3] Key Laboratory of Eco-Environment and Meteorology for the Qinling Mountains and Loess Plateau, Shaanxi Provincial
Meteorology Bureau, Xi'an 710015, Shaanxi, China
[4] State Key Laboratory of Soil Erosion and Dryland Farming on the Loess Plateau, Institute of Water and Soil Conservation,
Northwest A&F University, Yangling 712100, Shaanxi, China
*Correspondence to*: Jianqiang He (jianqiang_he@nwsuaf.edu.cn)





**Abstract.** Plastic film mulching has been extensively used to increase crop yields in arid and semi-arid regions, but it also altered agricultural landscapes and caused severe environmental pollution. Therefore, accurate and timely mapping of plastic-mulched farmland (PMF) distributions is crucial for planning agricultural production and preventing micro-plastic pollution. However, the scarcity of sufficient and representative training samples hinders large-scale supervised classification and extraction of PMF. Additionally, it remains unclear whether a pre-trained classifier can be directly applied to different regions and years for rapid PMF mapping. To address these challenges, we proposed a new framework that simultaneously takes advantages of the automation of index-based method and the generalization ability of supervised classifier-based approach for PMF mapping. Based on the distinctive spectral responses induced by plastic film deployment events, two novel and robust PMF indices—the Max Blue Band-based Plastic-mulched Farmland Index (MBPMFI) and the Blue Band-based Plastic-mulched Farmland Index (BPMFI)—were initially designed to automatically and rapidly extract PMF pixels in cloud-free areas as candidate training samples. Additionally, the transferability of classifiers pre-trained with these automatically generated samples and optimal features was further evaluated in spatial and spatial–temporal transferability scenarios using F1 values. Finally, by coupling the index-based training sample generation method with the temporal classifier transfer approach, PMF distributions were rapidly produced for the Loess Plateau of China (PMF-LP) for 2019–2021. The results showed that the two newly established indices, MBPMFI and BPMFI, were more robust than the existing PMF indices in enhancing PMF information and suppressing complicated backgrounds. The temporal classifier transfer proved suitable for directly and rapidly mapping PMF across multiple years without additional training samples. Using the locally adaptive classifiers as a reference, the average accuracy decrease of the transferred classifiers was less than 7.0% under the temporal transferability scenario. Our mapping framework achieved F1 values of 0.80–0.86 in recognizing PMF distributions for the Loess Plateau, highlighting its ability to delineate large-scale spatial patterns of PMF. Additionally, the estimated PMF areas based on the PMF-LP aligned well with the agricultural census data at municipal level ($R^2 > 0.87$). The framework developed in this study lays a foundation for future monitoring of PMF distributions and agricultural micro-plastic pollution on a large scale. The full archive of PMF-LP is freely available at https://doi.org/10.5281/zenodo.13369426 (Zhao et al., 2024).





## 1 Introduction


Plastic film mulching has been widely promoted and applied in China since 1978, due to its ability to
improve grain crop yields and water use efficiency through conserving water, maintaining soil moisture,
and increasing soil temperature (Liu et al., 2014; Sun et al., 2020; Zhao et al., 2023). Over the past 20 years,
China has consistently ranked first in the world in terms of the usage and area of plastic film (Yang et al.,
2015; Yan et al., 2014). The areas of plastic film mulching in China have increased from 11.0 million ha in
2000 to 17.6 million ha in 2020 (Zhang et al., 2022d). However, the extensive application of plastic films
has also caused severe environmental issues. Plastic films, mainly composed of polyvinyl chloride, always
have residues that are highly difficult to degrade in the soil, leading to severe "white pollution" (Liu et al.,
2014; Kumar et al., 2020; Gao et al., 2019). Furthermore, plastic film mulching might have impacts on the
regional climates, since the high-reflectivity and gas-tightness of the plastic film can alter the material and
energy exchange between the land surface and the atmosphere (Lu et al., 2014; Zhao et al., 2023). Therefore,
precise information about the spatial distributions of plastic-mulched farmland (PMF) over large areas is
essential for planning agriculture production, mitigating plastic residue pollution, and understanding water
and energy cycles in the agroecosystems (Veettil et al., 2023; Kumar et al., 2020).
Plastic film mulching data are usually derived from labor-intensive and time-consuming field surveys,
and then documented by the statistical yearbooks. However, statistical data often lack accurate information
about the locations and distributions of PMF. Benefiting from advances in Earth observation techniques,
the quantity and accessibility of remote sensing images have increased, enabling effective large-scale
monitoring of agriculture in a cost-effective and timely manner. (Maselli et al., 2020; Weiss et al., 2020;
Phiri et al., 2020). In particular, Sentinel-2 satellites launched by the European Space Agency (ESA),
provide data with a 10-m spatial resolution and a five-day revisit cycle, facilitating accurate monitoring of
the Earth's surface (Drusch et al., 2012; Chaves et al., 2020). Furthermore, the emergence of cloud-based
geospatial processing platforms, particularly the Google Earth Engine (GEE) (Gorelick et al., 2017), has
significantly enhanced the processing capabilities of remote sensing data (Tamiminia et al., 2020; Pham-
Duc et al., 2023). These advancements have supported the establishment of region- and nation-wide land
cover maps, such as the China Land Cover Dataset (CLCD) (Yang and Huang, 2021) for China, and the
Annual Crop Inventory (ACI) for Canada (Fisette et al., 2013). However, up to date, PMF has received
comparatively less attention than crops, and regional or national PMF maps are rarely publicly available,
particularly in China, where has the largest usage and area of plastic film.
Most optical satellite-based PMF mapping methods rely on spectral characteristics, as PMF can be
easily differentiated from other land cover types during the crop sowing period (Hasituya et al., 2016; Xiong
et al., 2019; Zheng et al., 2022). In this period, plastic films are applied to the surface of cultivated lands,
resulting in a distinctive bright-white color characteristic for PMF (Lu et al., 2014; Hasituya et al., 2020).
Thus, several studies have attempted to mapping PMF based on the unique color and spectral changes



caused by temporal variations in the "plastic film–vegetation–soil" composition (Lu et al., 2015; Xiong et al., 2019; Hao et al., 2019; Fu et al., 2022; Zhou et al., 2023). Additionally, spectral indices derived from images taken during the crop sowing period, such as the Plastic-mulched Landcover Index (PMLI) (Lu et al., 2014) and the Modified Plastic-mulched Cropland Index (mPMCI) (Fu et al., 2022), have been used to recognize PMF. Unfortunately, the time window for PMF recognition is very short (approximately one month) and mainly spans from April to May in northern China (Xiong et al., 2019; Cheng et al., 2023). As a result, obtaining seamless optical images over large areas during this period for index-based PMF mapping is challenging due to the inevitable cloud contamination.

Another approach for PMF mapping is machine learning, which exhibits less dependence on images obtained during the crop sowing periods. Owing to the predictive abilities of machine learning, even if a PMF pixel is cloud-contaminated during the crop sowing period, machine learning classifiers can still effectively identify it based on clear-sky images from other times (Zhang et al., 2022c; Wang et al., 2019; Gao et al., 2023). Thus, machine learning classifiers such as the Random Forest (RF) and Support Vector Machine (SVM) have been widely used for PMF mapping and achieved satisfactory accuracy (Hasituya and Chen, 2017; Lu et al., 2018; Zheng et al., 2022). Additionally, the predictive capabilities of machine learning classifiers combined with multi-temporal satellite observations also facilitated the seamless land cover mapping at regional and national scales (Zhang et al., 2022c; Zhang et al., 2021a; Liu and Zhang, 2023). However, implementing machine learning classifiers for large-scale land cover classification, particularly for PMF, remains a significant challenge due to the lack of abundant and representative training samples (Skakun et al., 2017; Foody and Arora, 1997; Wen et al., 2022).

To train machine learning classifiers, it is an efficient method to leverage existing land use/cover maps to obtain abundant training samples (Zhang et al., 2022b; Xuan et al., 2023; Zhang et al., 2022a; Zhang and Roy, 2017). Unfortunately, there are no publicly available PMF maps in China, which limits the application of this method. Since the index-based methods can rapidly and automatically identify specific land cover types in areas with high-quality image observations (Deng and Wu, 2012; Zhang et al., 2022c), some researchers have attempted to generate training samples for certain land cover types using index-based methods to address the lack of base maps for training sample generation. Thus, index-based approaches, combined with machine learning classifiers, have been gradually employed for automatic and seamless land cover classification (Zang et al., 2023; Zhang et al., 2022c; Yang et al., 2023). For example, Zhang et al. (2022c) demonstrated that integrating Winter Rapeseed Index-derived training samples with the RF classifier can seamlessly and automatically recognize rapeseed in large cloudy regions. For PMF recognition, most existing PMF indices were established in specific regions in Xinjiang Province, China (Lu et al., 2014; Xiong et al., 2019; Hao et al., 2019; Fu et al., 2022). However, different from the large and uniformly structured land parcels in Xinjiang, most regions in China typically have smaller and more scattered land parcels. Thus, the existing PMF indices struggle to distinguish PMF from the complex and fragmented backgrounds in these areas. Therefore, to fully leverage the strengths of index-based methods



and machine learning classifiers for automatic and seamless PMF mapping, a more robust PMF index is
urgently needed to separate PMF from other land cover types under diverse background conditions.
Moreover, classifier transfer, which involves applying classifiers trained in a source domain to
accomplish related tasks in a target domain, is also employed to address the scarcity of training samples
(Pan and Yang, 2009; Ma et al., 2024). This approach supports more rapid land cover mapping across large
scales and multiple years compare to locally adaptive classifiers. Consequently, it has been employed to
swiftly identify crop distributions in regions lacking ground reference samples and to retrace the spatial
distributions of historical crop types in the same region (Wang et al., 2019; Hu et al., 2022; Orynbaikyzy et
al., 2022; Qadir et al., 2024b; Qadir et al., 2024a). However, existing studies predominantly focused on
transferring supervised classifiers for crop classification. It remains unclear whether this approach is
applicable for the rapid mapping of PMF at large scales. And the factors influencing the accuracy of PMF
identification using this method are also unknown.
This study aimed to propose a novel framework for the rapid and automatic mapping of PMF
distributions across different regions and years at regional scales. Subsequently, we applied this framework
to mapping PMF distributions in the Loess Plateau of China from 2019 to 2021. The main objectives were:
(1) to automatically generate training samples based on newly proposed PMF indices that can robustly
enhance PMF information and suppress background noise under different background conditions; (2) to
assess the spatial–temporal transferability of the pre-trained classifiers for PMF mapping and identify the
factors that influence the transferability; (3) to rapidly and automatically produce seamless PMF
distribution maps (PMF-LP) for the Loess Plateau of China from 2019 to 2021 using the automatically
generated training samples and classifier transfer methods; and (4) to validate the accuracy of the PMF-LP
with independent ground samples and survey statistics.
**2  Study area and data**
**2.1 Study area**
This study identified PMF distributions from 2019 to 2021 in the Loess Plateau (33°43′ N–41°16′ N,
100°54′ E–114°33′ E) of China. The Loess Plateau covers an area of approximately $6.4 \times 10^5$ km$^2$, spanning
seven provinces: Qinghai, Gansu, Ningxia, Inner Mongolia, Shaanxi, Shanxi, and Henan (Fig. 1 (a)). This
region features diverse topography, with elevations ranging from about 300 meters above sea level in the
Guanzhong Plain to 5,000 meters in Qinghai Province (Fig. 1 (b)). According to the Köppen-Geiger climate
classification (Beck et al., 2023), the Loess Plateau experiences a warm temperate continental monsoon
climate, characterized by hot-rainy summers, and cold-dry winters (Fig. 1 (c)). The mean annual
precipitation varies across the region, from 200 mm in the northwest to 800 mm in the southeast, with 60–
70% of the total precipitation falling between July and October. The annual evapotranspiration ranges from
1,400 to 2,000 mm, and the average annual temperature varies between 3.6℃ and 14.3℃ (Li et al., 2024;
Tang et al., 2018; Li et al., 2012). Plastic-film mulching, an effective field management practice for storing

and maintaining soil moisture, is widely used across the Loess Plateau (Fig. 1 (d)~(g)) (Zhang et al., 2022f; Wang et al., 2023). Notably, transparent plastic film predominates in this region, covering about 90% of the total plastic-mulched fields (Lu et al., 2014; Wang et al., 2016; Hasituya et al., 2020). Thus, this study focused solely on farmland mulched with transparent plastic film. To balance the computational cost and the need for adequate training samples, the study area was divided into 32 mapping units based on the municipal administrative boundaries (Fig. S1).

Three major crops—maize, potato, and winter wheat—are extensively cultivated in the Loess Plateau. Maize and potato, the primary plastic-mulched crops, are typically sown in mid-April and harvested in late September, with plastic film applied in a short time before and after sowing. Winter wheat is sown in October of the prior year and harvested in late June. Only a very small portion of winter wheat is mulched with plastic film in the study area. The crop calendars for these main crops were illustrated in Table 1.

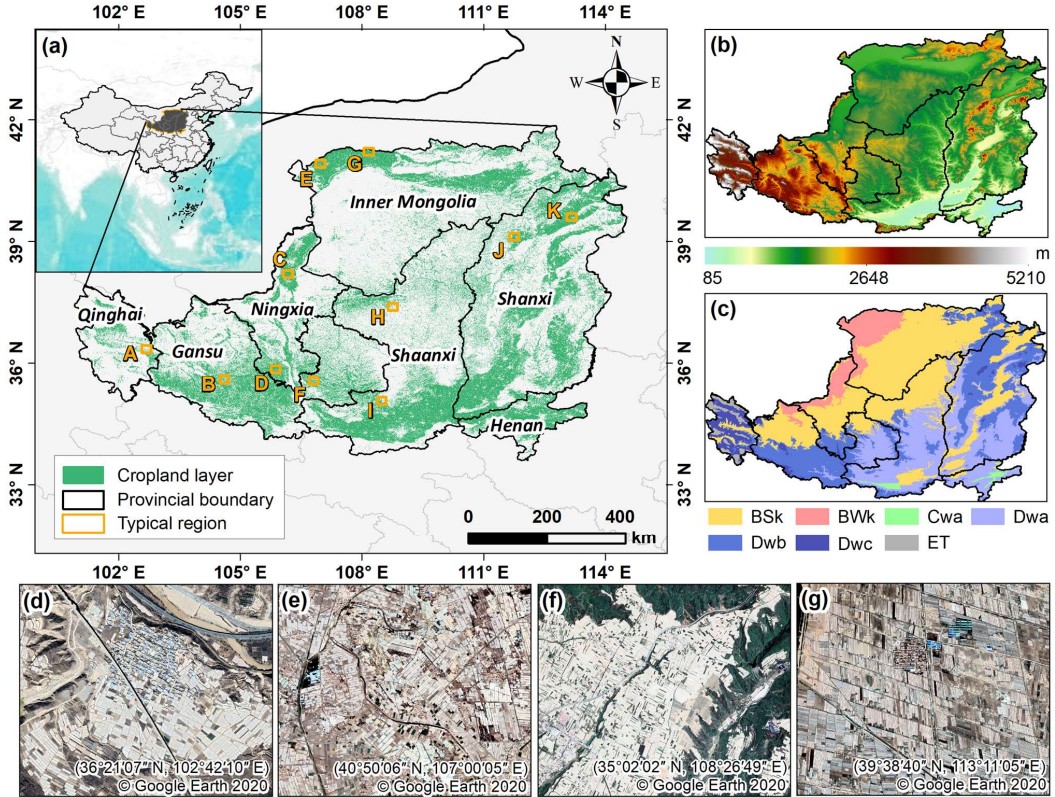

**Figure 1.** Overview of the study area. (a) Location of the Loess Plateau in China, spatial distributions of croplands. Rectangular regions (A~K) are representative mulched regions where high-resolution Google Earth images are available for visual intepretation. (b) Topography across the Loess Plateau. Digital Elevation Model (DEM) with a spatial resolution of 30 m was obtained from the Shuttle Radar Topography Mission (SRTM) (Farr et al., 2007). (c) Köppen-Geiger climate classification in the Loess Plateau. BSk: arid, steppe, cold; BWk: arid, desert, cold; Cwa: temperate, dry winter, hot summer; Dwa: cold, dry winter, hot summer; Dwb: cold, dry winter, warm summer; Dwc: cold, dry winter, cold summer; and ET: polar, tundra. (d)~(g) Zoom-in views of Google Earth images of plastic-mulched farmland during the mulching stage in 2020 for the rectangular regions of A, E, I, and K, resepctively.



**Table 1.**
Crop calendars illustrating growing phases of three major crops (i.e., maize, potato, and wheat) in the Loess Plateau of
China. The labels "E", "M", and "L" denote the early, middle, and last 10-day phases of a month.

| Crops | Jan | | | Feb | | | Mar | | | Apr | | | May | | | Jun | | | Jul | | | Aug | | | Sep | | | Oct | | | Nov | | | Dec | | |
|---|---|---|---|---|---|---|---|---|---|---|---|---|---|---|---|---|---|---|---|---|---|---|---|---|---|---|---|---|---|---|---|---|---|---|---|---|
| (E/M/L) | E | M | L | E | M | L | E | M | L | E | M | L | E | M | L | E | M | L | E | M | L | E | M | L | E | M | L | E | M | L | E | M | L | E | M | L |
| Maize | | | | | | | | | | | | Sowing | | Seeding | | | Jointing | | | Heading | | | Milk-ripe | | | Maturity | | | | | | | | | | | |
| Potato | | | | | | | | | | | | | Sowing | | Seeding | | | | | Flowing | | | | | Tuber-swelling | | Maturity | | | | | | | | | |
| Wheat | Seeding | | | | | | Jointing | | | | | Earing | | Maturity | | | | | | | | | | | | | | Sowing | | | | | Seeding | | | |


## 2.2 Data

### 2.2.1 Sentinel-2 data

The Sentinel-2 satellite, equipped with multispectral instruments, has 13 spectral bands covering
visible (10 m/pixel), near-infrared (20 m/pixel), and shortwave-infrared (20 m/pixel) wavelengths (Han et
al., 2021; Drusch et al., 2012). Its 5-day revisit cycle at the equator provides high-frequency Earth
observation data for monitoring rapid changes on the Earth's surface (Zhang et al., 2020b). In this study,
all available Sentinel-2 surface reflectance (SR) data covering the Loess Plateau from 2019 to 2021,
archived in GEE, were employed as input data since SR data are unavailable for this region before 2019.
The Sentinel-2 SR products in GEE platform have been pre-processed with radiometric and atmospheric
corrections. Considering the growth patterns of plastic-mulched crops (Table 1), Sentinel-2 data were
further limited to March to October, and quality assessment (QA) bands were used to mask clouds and
cirrus pixels.
Given that at least five points were required to fit a second-order harmonic regression curve whose
coefficients served as the input variables for the classifiers (Section 3.2), at least five images per pixel were
needed. We counted the number of yearly cloud-free images per pixel from 2019 to 2021 (Fig. S2), and
found that nearly 100% of the pixels in the study area had more than five observations each year. The mean
numbers of cloud-free images per pixel were 46 in 2019, 45 in 2020, and 43 in 2021, respectively. Therefore,
the Sentinel-2 time series images provided ample data for the image processing requirements of this study.

### 2.2.2 Cropland mask

To simplify the process of PMF mapping and reduce commission errors caused by other land cover
types (Zhang et al., 2023; You et al., 2023; You et al., 2021; Defourny et al., 2019), two cropland layers,
including the ESA WorldCover 2020 (Zanaga et al., 2022) and the China Land Cover Dataset (CLCD)
(Yang and Huang, 2021) of 2020, were overlapped and utilized to exclude non-cropland pixels in this study.
The idea to incorporate two different cropland layers was driven by the assumption that relying on a single
cropland layer could introduce uncertainties, while combining two layers could generate cropland pixels
with a higher confidence level (You et al., 2023). Since the CLCD has a spatial resolution of 30 m, we re-
sampled it to a 10-m resolution using the nearest-neighbor method. Given the relatively stable nature of

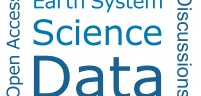

cropland across adjacent years (Gong et al., 2019), we applied the overlapped cropland layer in 2020 to
mask Sentinel-2 images for each year from 2019 to 2021.

### 2.2.3 Google Earth high-resolution images (GE-HRIs)

Google Earth high-resolution images, sourced from the Google Earth platform, provide satellite and
aerial imagery of the Earth's surface with a sub-meter spatial resolution of 0.6 m, which facilitates sample
collection through visual interpretation. Therefore, the GE-HRIs were utilized to assist in obtaining
validation samples in this study. Additionally, two representative rectangular regions (20 km × 20 km) with
at least one GE-HRI image available during the mulching stage were selected in each province (yellow
areas A~K in Fig. 1(a)). The samples interpreted in these rectangular regions (300 PMF/Non-PMF points
for each region) were employed to determine optimal thresholds for the newly proposed PMF indices in
Section 3.1.2.

### 2.2.4 Agricultural statistics

The PMF census data from 2019 to 2021, sourced from municipal-level agricultural statistical offices,
were collected to assess the mapped PMF areas. It is noteworthy that plastic film usage in the statistical
yearbooks of most cities in the Loess Plateau is quantified in tons. Consequently, in this study, we only
obtained statistical data for 12 cities (Fig. S3), where plastic film usage was measured in terms of area.
Since these 12 cities account for nearly one-third of the total area of the Loess Plateau, the accuracy
assessment of the mapped PMF areas based on the municipal statistical data was considered representative
and reliable.

## 3  Methodology

The workflow of this study was summarized in Fig. 2. First, cloud-free Sentinel-2 time series data
were used to fit harmonic curves, generating feature variables that served as input variables for the
classifiers. Next, a temporal signature analysis of various cropland-related land covers was conducted.
Distinctive signatures during the mulching period guided the design of PMF indices, which effectively
separated PMF from other land cover types. These indices were then employed to identify PMF in cloud-
free areas and automatically generate training samples. Third, random forest classifiers were developed to
recognize PMF based on the feature variables and the automatically generated training samples. The
transferability of these classifiers was then evaluated in terms of temporal and spatial–temporal
transferability. Finally, the PMF mapping results were assessed against ground reference samples and
statistical PMF area data.



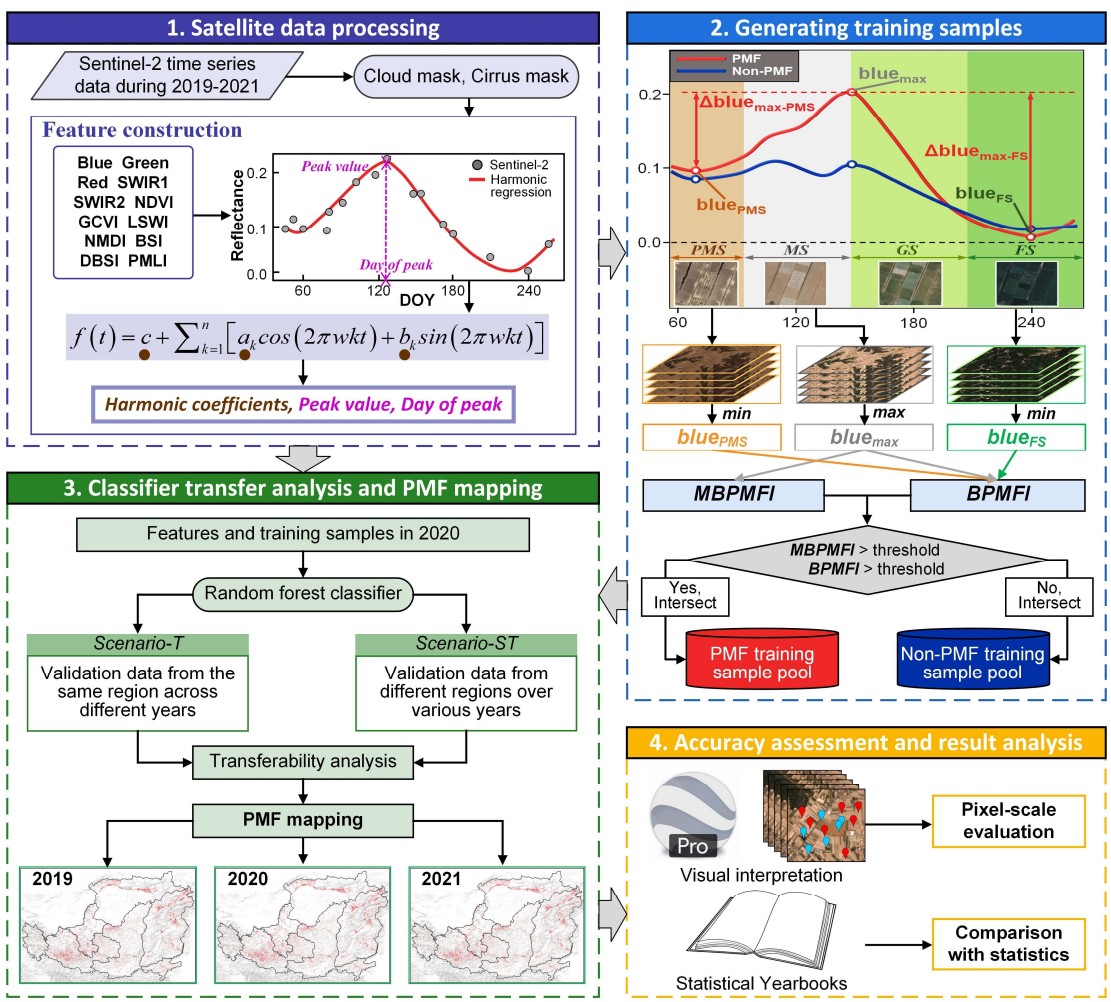

**Figure 2.** Flowchart depicting the integration of index-based training sample generation and classifier transfer for plastic-mulched farmland mapping. PMF: Plastic-mulched Farmland, Non-PMF: Non-plastic-mulched Farmland. The entire growing season of plastic-mulched crops was divided into four stages: pre-mulching stage (PMS), mulching stage (MS), growing stage (GS), and flourishing stage (FS) (Section 3.1.2). The MBPMFI (Max Blue Band-based Plastic-mulched Farmland Index) and BPMFI (Blue Band-based Plastic-mulched Farmland Index) are the two novel PMF indices proposed in this study (Section 3.1.2).

## 3.1 Automatically generating training samples

### 3.1.1 Optical characteristics of the PMF

The multi-temporal Sentinel-2 images with true-color composites from a representative sub-region in Pingliang City, Gansu Province, were shown in Fig. 3 (c)~(j) to analyze the optical characteristics of PMF. The PMF, plastic greenhouses (PGs), and vegetation (mainly winter crop) within this sub-region represented the primary agricultural-related land cover types on the cultivated farmland in the Loess Plateau. During the crop sowing period, PMF exhibited a bright-white color due to the plastic films, making it easily





distinguishable from vegetation (dark-green color) but potentially confusing it with PGs (blue-gray or white color) (Fig. 3 (c)~(f)). Subsequently, from June to September (Fig. 3 (g)~(i)), the crop canopy covered the PMF, giving it a green appearance similar to vegetation. Meanwhile, since crops grew inside the PGs, the PGs maintained blue-gray or white appearance consistently. Thus, this period was ideal for differentiating between PGs and PMF.

Based on this analysis, we observed that PMF exhibited a bright-white appearance during the crop sowing period. Theoretically, the conspicuous bright white characteristic indicated the substantial reflectance of PMF in the visible spectrum. Temporal profiles of the Sentinel-2 blue band reveled distinguishable patterns among PMF, PGs, and vegetation (Fig. 3 (b)). Specifically, during the crop sowing period, areas covered by plastic films had higher reflectance in the blue band compared to those covered by the crop canopy. Subsequently, as crops grew in the PMF, the increased absorption in the visible spectrum led to a gradual decrease in PMF reflectance in the blue band. Particularly, during the peak growing stage of crops, PMF reached its lowest reflectance in the blue band.

In addition to the blue band, PMF demonstrated clear differentiation from other land cover types in the green and red bands, as well as in the red-edge band (red edge1) (Fig. S4). Notably, this study focused exclusively on the blue band due to the consistent temporal profiles observed across these four bands. Furthermore, the distinct PMF characteristics in the blue band were also tested across different regions in the Loess Plateau (Fig. S5). The results showed that PMF reflectance in the blue band exhibited the same temporal profiles as those illustrated in Fig. 3 (b), characterized by a rapid increase during the crop sowing period, followed by a swift decrease during the flourishing period.

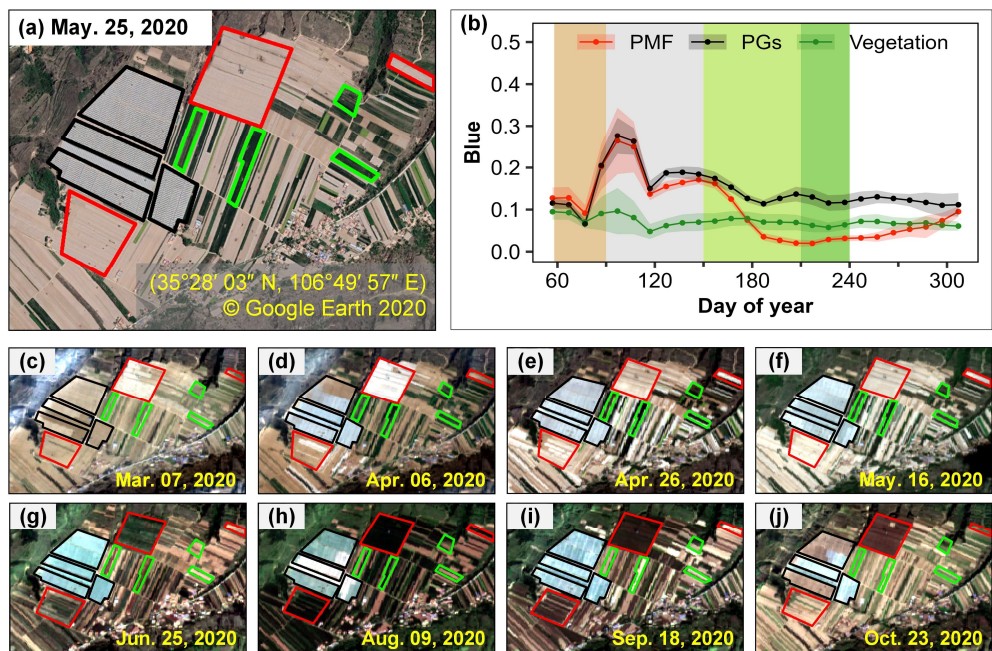

**Figure 3.** Time-series profiles of plastic-mulched farmland (PMF), plastic greenhouses (PGs), and vegetation in a representative sub-region of Pingliang City, Gansu Province, China. (a) High-resolution Google Earth imagery during the crop sowing period. The red, black, and green outlined plots denote the PMF, PGs, and vegetation, respectively. (b) Time series of PMF, PGs, and vegetation in the Sentinel-2 blue band. The red, gray, and blue buffers indicate one standard deviation. The brown, gray, yellowish-green, and green rectangular areas denote the pre-mulching stage (PMS), mulching stage (MS), growing stage (GS), and flourishing stage (FS) as defined in Section 3.1.2, respectively. (c)~(j) Sentinel-2 images with a true-color composite of blue, green, and red bands on different dates across the entire crop growing season.

### 3.1.2 Developing PMF indices to automatically generate training samples

Following the temporal profile analyses in Section 3.1.1, the entire growing season of plastic-mulched crops was divided into four distinct stages (Fig. 4). The pre-mulching stage (PMS) (March to early April) involves primarily field plowing, with seldom plastic film coverage on the farmland. The mulching stage (MS) (mid-April to May) represents the gradual coverage of farmland with plastic films. During the growing stage (GS) (June to July), crops begin to grow but do not yet fully cover the cropland. The flourishing stage (FS) (August to September) is characterized by vigorous crop growth, with the crop canopy fully sheltering the cropland.

Across these stages, the distinctive spectral responses of PMF in the blue band were summarized as follows: a relatively large range of reflectance dynamics from PMS to MS ($\Delta blue_{max-PMS}$) due to the gradual deployment of plastic film, and the maximal range of reflectance dynamics from GS to FS ($\Delta blue_{max-FS}$) due to the abundant visible spectrum absorption for photosynthesis. Considering that both $\Delta blue_{max-PMS}$ and $\Delta blue_{max-FS}$ for PMF were relatively higher than those for other land cover types, multiplying these two metrics could effectively highlight PMF while suppressing other land cover types. Specifically, two



local troughs ($blue_{PMS}$ and $blue_{FS}$) are the minimum reflectance values of the blue band during the PMS and
FS, respectively. The local peak ($blue_{max}$) refers to the maximum reflectance value of the blue band during
the MS. Additionally, the metric $blue_{max}$ was also employed to enhance PMF signals, as it exhibited the
highest reflectance value throughout the period from PMS to FS. Finally, two PMF indices, the Max Blue
Band-based Plastic-mulched Farmland Index (MBPMFI) (Eq. (1)) and the Blue Band-based Plastic-
mulched Farmland Index (BPMFI) (Eq. (2)) were defined.
$$MBPMFI = blue_{max} \tag{1}$$
$$BPMFI = 100 \times \Delta blue_{max\text{-}PMS} \times \Delta blue_{max\text{-}FS} = 100 \times (blue_{max} - blue_{PMS}) \times (blue_{max} - blue_{FS}) \tag{2}$$
where the multiplier 100 was used to linearly stretch the BPMFI to a wider range.

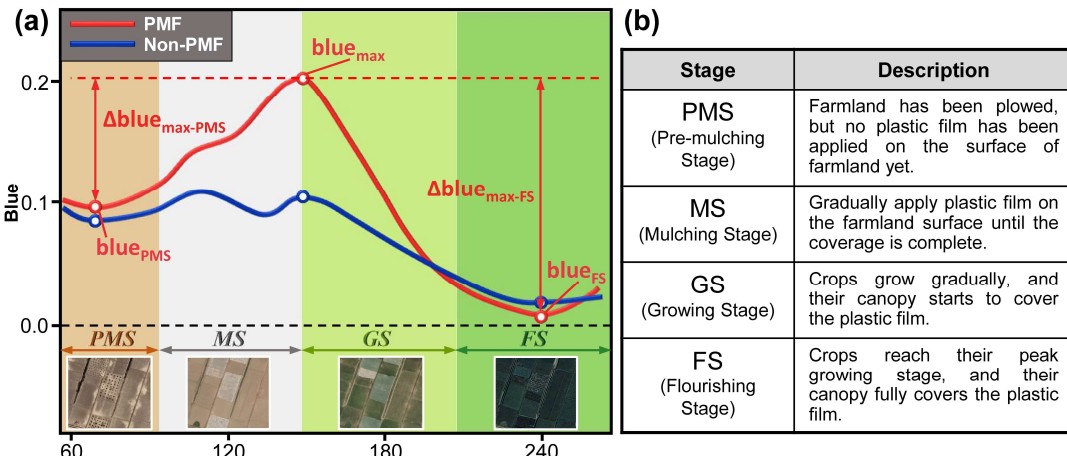

| Stage | Description |
|---|---|
| PMS (Pre-mulching Stage) | Farmland has been plowed, but no plastic film has been applied on the surface of farmland yet. |
| MS (Mulching Stage) | Gradually apply plastic film on the farmland surface until the coverage is complete. |
| GS (Growing Stage) | Crops grow gradually, and their canopy starts to cover the plastic film. |
| FS (Flourishing Stage) | Crops reach their peak growing stage, and their canopy fully covers the plastic film. |

**(c)**

| Value | Description |
|---|---|
| $blue_{PMS}$ | This value, calculated using the minimum composite method during the PMS, represents the local trough in this stage. |
| $blue_{max}$ | This value, calculated using the maximum composite method during the MS, represents the maximum reflectance across the entire period from PMS to FS. |
| $blue_{FS}$ | This value, calculated using the minimum composite method during the FS, represents the local trough caused by the absorption of visible spectrum by the crops in this stage. |
| $\Delta blue_{max\text{-}PMS}$ | This value reflects the relatively large range in the reflectance dynamics from PMS to MS, attributed to the gradual deployment of plastic film on the surface of farmland. |
| $\Delta blue_{max\text{-}FS}$ | This value reflects the maximum range in the reflectance dynamics from GS to FS, resulting from the abundant absorption of visible spectrum for photosynthesis. |


**Figure 4.**    Schematic diagrams illustrating the construction of the plastic-mulched farmland indices. (a) Time-series
profiles of the plastic-mulched farmland (PMF) and non-plastic-mulched farmland (Non-PMF) in the Sentinel-2 blue band.
(b) Defined stages used to divide the entire growing season of plastic-mulched crops into four parts. (c) Explanation of the
variables used to represent the unique characteristics of PMF.



Notably, since the primary objective of this study was to recognize PMF and given the relative scarcity
of PGs compared with PMF (Feng et al., 2021; Tong et al., 2024), land cover types were categorized into
two classes: PMF and Non-PMF. Table S1 presented the statistical analysis results of the MBPMFI and
BPMFI indices for both PMF and Non-PMF samples collected from the rectangular regions (Section 2.2.3).
Generally, the *p-values* for the two PMF indices across different provinces were equal to 0, indicating the
great potential of MBPMFI and BPMFI to enhance PMF signals while suppressing Non-PMF signals.
Referring to previous studies (Zhang et al., 2022g; Zhou et al., 2024), sample points from the rectangular
regions were exploited to establish province-specific thresholds for each PMF index. The optimal
thresholds for the two novel indices were separately determined based on accuracy information with
different thresholds (Fig. 5). Particularly, for Henan Province, we adopted the same thresholds as those used
in Shanxi Province. Finally, PMF and Non-PMF pixels in cloud-free areas were automatically extracted.
To improve the reliability of the training samples, only cropland classified as PMF or Non-PMF by both
PMF indices was included in the final training sample pools (Fig. 6).

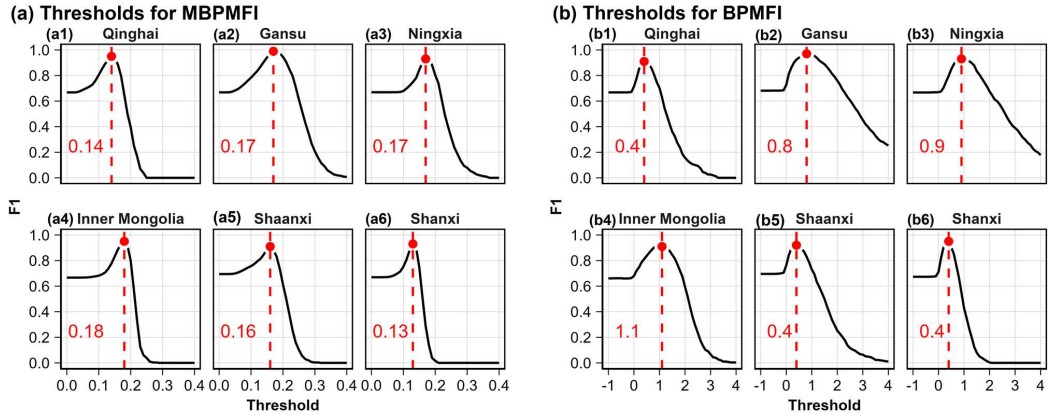


**Figure 5.**   Classification accuracy curves of various plastic-mulched farmland indices at different thresholds.

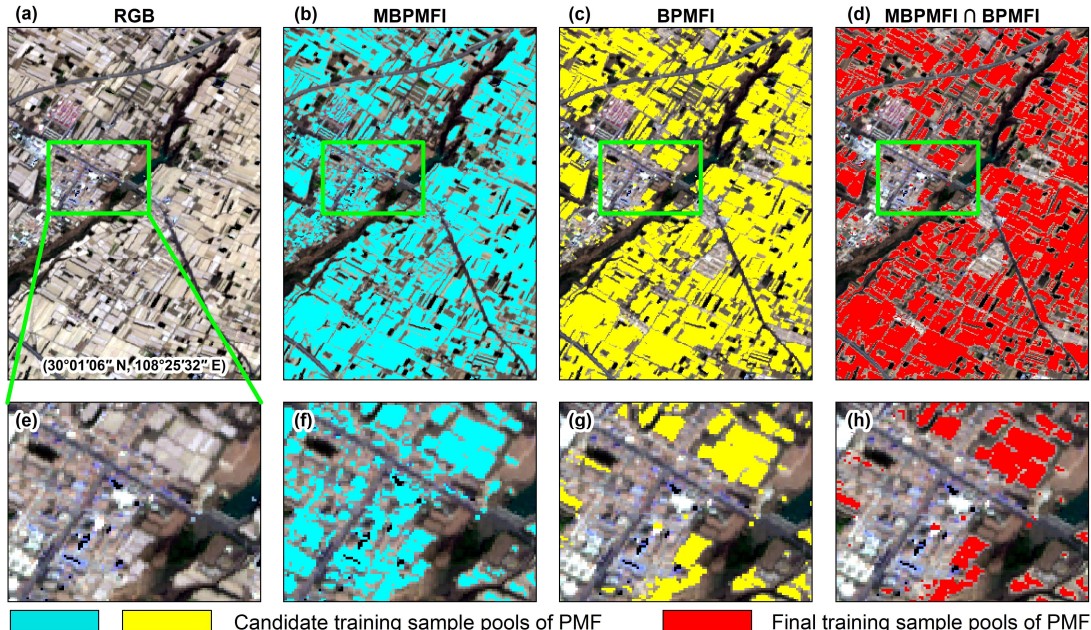

**Figure 6.** Zoomed-in views for illustrating the detailed process of training sample generation. (a) Sentinel-2 images with a true-color composite of blue, green, and red bands. (b) Candidate training sample pool derived based on MBPMFI (Max Blue Band-based Plastic-mulched Farmland Index). (c) Candidate training sample pool derived based on BPMFI (Blue Band-based Plastic-mulched Farmland Index). (d) Final training sample pool. "∩" means the intersection is implemented for sample pools of MBPMFI and BPMFI. (e)~(h) Zoomed-in views from the green frame in (a) for presenting the misclassification between plastic-mulched farmland (PMF) and impervious surfaces.

### 3.1.3 Training sample generation and refinement

Based on the candidate sample pools generated in Section 3.1.2, a stratified random sampling approach was adopted to select 2000 pixels for PMF and Non-PMF within every mapping unit for the year 2020. To further ensure the reliability of these samples and eliminate potential errors, a strict spatial filter (8-neighbor filter) was conducted for these samples (Zhang et al., 2022a; Zhang et al., 2023; Zhang and Roy, 2017). Only pixels sharing the same class type as the surrounding eight pixels were considered as high-quality samples. These samples were utilized as training data for subsequent classifiers to identify PMF.

### 3.2 Feature calculation and selection

The visible bands (Blue, Green, and Red) and shortwave-infrared bands (SWIR1 and SWIR2) from Sentinel-2 were selected as input features for classifiers, due to their high potential to separate PMF from other land cover types (Hasituya et al., 2016; Xiong et al., 2019; Hao et al., 2019). Since plastic films can influence energy balance and water cycles on the land surface (Lu et al., 2014; Hasituya and Chen, 2017), several remote-sensing vegetation indices associated with vegetation and soil conditions were also included as input features (Table S2): NDVI (Tucker, 1979), GCVI (Gitelson et al., 2005), LSWI (Xiao et al., 2002), NMDI (Wang and Qu, 2007), BSI (Rikimaru et al., 2002), DBSI (Rasul et al., 2018), PMLI (Lu et al., 2014).



Moreover, the newly proposed PMF indices, MBPMFI and PMFI, were also incorporated.
Compared with Non-PMF, PMF exhibited unique temporal profiles resembling sine waves in the blue
band (Fig. 4 (a), Fig. S5). To characterize these profiles and fill missing values caused by cloud
contaminations, harmonic regression (Jakubauskas et al., 2001; Zhou et al., 2022) (Eq. (3)) was conducted
to fit time series curves for PMF across the five surface reflectance bands and seven vegetation indices. The
time window for fitting the time-series curves extended from March 1st to October 31st, covering the entire
growing season of plastic-mulched crops. Each band was treated as a time-dependent function, denoted as
f(t).

$$f(t) = c + \sum_{k=1}^{n} \left[ a_k \cos(2\pi k\omega t) + b_k \sin(2\pi k\omega t) \right] \tag{3}$$

where the independent variable t is the day of year expressed as a fraction between 0 (January 1st) and 1
(December 31st) for a satellite image, c is the intercept term, n is the order of harmonic series, $a_k$ are the
cosine coefficients, $b_k$ are the sine coefficients, and $\omega$ is the angular frequency.
In the above harmonic regression formula, parameters n and $\omega$ need to be adjusted to balance the fitting
closeness to the observation points and prevent overfitting. Based on the ground reference PMF samples
within the rectangular regions (yellow areas A~K in Fig. 1 (a)), we picked n = 2 and $\omega$ = 1.5 as the optimal
parameters by evaluating the root mean square error (RMSE) of the fitting progress (Fig. S6). Therefore,
we obtained the final harmonic regression formula (Eq. (4)).

$$f(t) = c + a_1 \cos(3\pi t) + b_1 \sin(3\pi t) + a_2 \cos(6\pi t) + b_2 \sin(6\pi t) \tag{4}$$

After the harmonic regression, the time series of each band was represented by five coefficients: c, $a_1$, $b_1$,
$a_2$, and $b_2$. Furthermore, the peak value (peak) of each band and their corresponding dates (timing) were
also extracted from the fitted harmonic regression curves. These coefficients combined with MBPMFI and
BPMFI resulted in a total of 86 (12 × 7 + 2 = 86) features.
To select the important features and discard unimportant ones which may adversely affected the
accuracy and computational cost of supervised classifiers (Zou et al., 2015; Wang et al., 2022b), a
combination of time-series correlation analysis and random forest feature importance analysis was
employed to reduce the number of features (Fig. S7). The reduction process included: (1) grouping bands
with correlation coefficients exceeding 0.90, retaining only the band with the highest feature importance
(retained bands: Blue, SWIR2, BSI, DBSI, GCVI, NMDI, and PMLI); (2) further limiting the seven
harmonic coefficients to the peak, $a_1$, and $b_2$ terms based on their importance evaluation results; (3) retaining
MBPMFI and BPMFI due to their higher importance compared with other features (Fig. 7). Ultimately, a
total of 23 features were adopted as input variables for the classifiers in this study. A comparison of PMF
identification accuracies using 86 features versus 23 features revealed no significant difference ($\alpha$ = 0.05)
in classification accuracy between the two sets of features (Fig. S8).

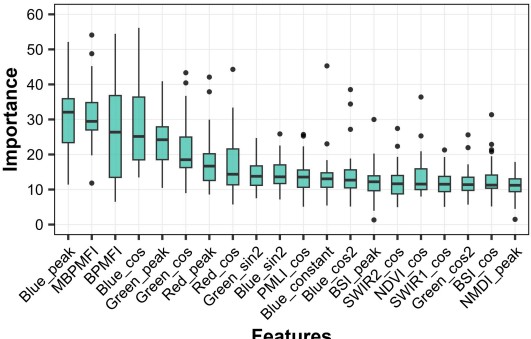


**Figure 7.** Feature importance of top 20 features calculated by the random forest classifiers trained in each city in the Loess Plateau. Since the feature importance in each city is not comparable, we normalized the individual value by dividing the total importance of all features in each city and scaled them by multiplying 1000.

### 3.3 Classifier transferability analysis

We employed the pixel-based random forest (RF) classifier (Breiman, 2001) available on the GEE platform to distinguish PMF from Non-PMF for each mapping unit, based on the selected optimal features and automatically generated samples mentioned above. As an ensemble algorithm comprising numerous decision trees, RF has been widely used for large-scale land cover classification (Zhang et al., 2021c; Liu et al., 2020), crop type mapping (Yang et al., 2023; Wang et al., 2020), and classifier transferability analysis (Wijesingha et al., 2024; Qadir et al., 2024b), due to its capabilities in handling high-dimensional data, tolerating noise, and preventing overfitting (Belgiu and Drăguţ, 2016; Sheykhmousa et al., 2020). Two key hyperparameters need to be set when using RF, which are the number of trees (*Ntree*) to be generated and the number of features(*Mtry*) used for testing the best split when growing the trees (Belgiu and Drăguţ, 2016). The former parameter was set as 100 since the out-of-bag errors in different mapping units ceased to decrease beyond 100 trees (Fig. S9); the latter one was set as the default value (i.e., the square root of the number of features).

Revealing the temporal and spatial–temporal transferability of classifiers can facilitate multi-year PMF mapping without the need to collect domain- and year-specific training samples. The classifiers trained with the optimal features and automatically generated samples in each mapping unit in 2020 (reference scenario, Scenario–Ref) were tested in two transferability scenarios to evaluate their temporal and spatial–temporal transferability: (1) classifiers trained in each mapping unit in 2020 were applied to recognize PMF within the same units for 2019 and 2021 to assess their temporal transferability (Scenario–T); and (2) classifiers trained in each mapping unit in 2020 were employed to recognize PMF in all other remaining units for 2019 and 2021 to test their spatial–temporal transferability (Scenario–ST). Note that classifiers trained with data from 2020 were not assessed in other spatial regions in the same year of 2020 because the primary goal of classifier transfer in this study is to rapidly retrace historical PMF distributions. Additionally, we trained classifiers in each mapping unit (municipal level) because sample point collection in practice is usually



conducted within administrative boundaries rather than in agro-ecological boundaries (Hao et al., 2020;
Wang et al., 2019). In each scenario, the F1-score (F1) was calculated based on validation samples as
accuracy evaluation metric. Furthermore, to assess changes of F1 in each transferability scenario against
reference scenario, a percentage change of F1 values ($F1_{change}$) was also computed (Eq. (5)).
$$F1_{change} = \frac{F1_{transferability} - F1_{reference}}{F1_{reference}} \times 100 \qquad (5)$$

**3.4 Accuracy assessment**
We assessed the quality of the generate PMF maps with two measures. First, we performed a pixel-
wise quantitative assessment for the PMF maps in 2019–2021, using more than 5000 ground reference
samples each year (Table S3). Four accuracy metrics were adopted: producer accuracy (PA), user accuracy
(UA), overall accuracy (OA), and F1-score (F1). Next, the coefficient of determination ($R^2$) was used to
quantitatively measure the consistency of the PMF areas derived from agricultural statistics and those from
the resultant maps. Due to the lack of publicly available PMF products in China, inter-comparison with
existing products was not possible in this study.
The validation samples were collected by visually interpreting GE-HRIs and Sentinel-2 images, based
on the unique characteristics of PMF (Section 3.1.1). A two-step strategy was designed to generate the
validation samples: (1) croplands exhibiting a white hue during the mulching stage (MS) and a dark green
hue during the flourishing stage (FS) in the true-color composite images were more likely to be PMF than
those showing other hue changes; (2) the time-series curves of PMF in the blue band displayed more
pronounced peak values during the mulching stage (MS) and trough values during the flourishing stage (FS)
compared to other land cover types. Pixels meeting above two rules were finally labeled as PMF; otherwise,
they were grouped into Non-PMF. Ultimately, we obtained 7,091, 12,140, and 6,714 validation samples for
2019, 2020, and 2021, respectively (Table S3, Fig. S10).
Moreover, five additional PMF indices (Table 2), including PMLI (Lu et al., 2014) (Eq. (6)), $PMFI_{first}$
(Xiong et al., 2019) (Eq. (7)), $PMFI_{second}$ (Xiong et al., 2019) (Eq. (8)), $PMLI_{SWIR}$ (Hao et al., 2019) (Eq.
(9)), and mPMCI (Fu et al., 2022) (Eq. (10)), were utilized to compare with the newly established MBPMFI
and BPMFI indices in enhancing PMF information while suppressing Non-PMF information. Except for
the PMLI, higher values of these indices are more related to PMF. To test the robustness of these indices,
three representative regions with diverse environmental conditions were selected: (1) Region I, located in
Bayan Nur City, Inner Mongolia Province, exhibits a large and densely distributed PMF area in this region
(Fig. 8 (a)); (2) Region II, located in Linxia City, Gansu Province, features an irregularly shaped PMF with
sparse distribution (Fig. 9 (a)); and (3) Region III is located in Xianyang City, Shaanxi Province, where
some plastic greenhouses and buildings exhibit similar spectral profiles to PMF (Fig. 10 (a)). A visual
evaluation was conducted to intuitively compare the performance of these indices. Since the presence of
noises in the index images could affect their display and subsequently influence the visual comparison





results, all index images were displayed with a 1% linear stretch (Zhang et al., 2022g) to ensure the fairness
of visual comparison among them.
**Table 2.**
Summary of the existing plastic-mulched farmland indices used for comparison with the newly established MBPMFI and
BPMFI indices in this study.

| Name | Equation* | Equation index |
|---|---|---|
| PMLI <br> (Plastic-mulched Landcover Index) | $\dfrac{\rho_{SWIR1} - \rho_{Red}}{\rho_{SWIR1} + \rho_{Red}}$ | (6) |
| PMFI$_{first}$ <br> (Plastic-mulched Farmland Index First) | $\dfrac{\rho_{SWIR2}}{\rho_{NIR}}$ | (7) |
| PMFI$_{second}$ <br> (Plastic-mulched Farmland Index Second) | $\dfrac{\rho_{SWIR2}}{\rho_{Blue}}$ | (8) |
| PMLI$_{SWIR}$ <br> (Plastic-mulched Land Index with Shortwave-infrared Band) | $\dfrac{(\rho_{RE4} + \rho_{NIR} + \rho_{RE3}) - (\rho_{SWIR1} + \rho_{SWIR2})}{\rho_{SWIR1} + \rho_{SWIR2}}$ | (9) |
| mPMCI <br> (Modified Plastic-mulched Cropland Index) | $\dfrac{\rho_{SWIR1} + \rho_{NIR}}{\rho_{SWIR1} - \rho_{NIR}}$ | (10) |

* $\rho_{blue}$, $\rho_{red}$, $\rho_{nir}$, $\rho_{RE3}$, $\rho_{RE4}$, $\rho_{SWIR1}$, and $\rho_{SWIR2}$ represent the Sentinel-2 reflectance in the blue band (496 nm), red band
(665 nm), near-infrared band (833 nm), red-edge 3 band (779 mm), red-edge 4 band (864 mm), shortwave-infrared band
1 (1610 mm), and shortwave-infrared band 2 (2185 mm), respectively.

## 4    Results

### 4.1 Comparisons with existing PMF indices

In this section, we presented detailed visual comparison results for all the seven PMF indices under
three different environmental conditions. High values of the PMFI$_{first}$, PMFI$_{second}$, PMLI$_{swir}$, mPMCI,
MBPMFI, and BPMFI correspond to PMF objects, while a high value of the PMLI is related to Non-PMF
objects. For the convenience of visual comparisons between the PMLI and the other six indices, the color
bar of PMLI was reversed. Consequently, in all seven index images, the red color is associated with PMF,
and the blue color is related to Non-PMF.

### 4.1.1 Visual comparison in Region I

In Region I, the MBPMFI and BPMFI indices outperformed the other five indices in highlighting PMF
and suppressing background noise (Fig. 8 (g) and (h)). The mPMCI performed the poorest, barely
recognizing PMF (Fig. 8 (f)). PMLI and PMLI$_{SWIR}$ showed similar performance, both misclassifying
vegetation as PMF. Region I is located in the Hetao Irrigation District of Inner Mongolia, where croplands
are extensively irrigated through the integral irrigation systems. The shortwave-infrared spectrum is highly
absorbed by water. Although PMF had a higher reflectance value in the red spectrum than vegetation, the
wet soil caused by irrigation can lead to lower reflectance values in shortwave-infrared spectrum for
vegetation than for PMF. In this context, both PMF and vegetation might exhibit a low reflectance value in
PMLI (Eq. (6)), leading to misclassification between them (Zhang et al., 2022g). While $PMLI_{SWIR}$ (Eq. (9))
was primarily designed to separate PMF from bare land, it failed to suppress vegetation, which also had a
high reflectance value in the near-infrared spectrum and low reflectance value in the shortwave-infrared
spectrum similar to PMF (Fig. 8 (e)). $PMFI_{first}$ and $PMFI_{second}$ could highlight PMF to a certain extent, but
wrongly highlighted some background objects, such as roads and bare lands (Fig. 8 (c) and (d)). Overall,
MBPMFI and BPMFI showed comparable performance in highlighting PMF signals, with BPMFI having
an advantage in suppressing background noise, especially roads.

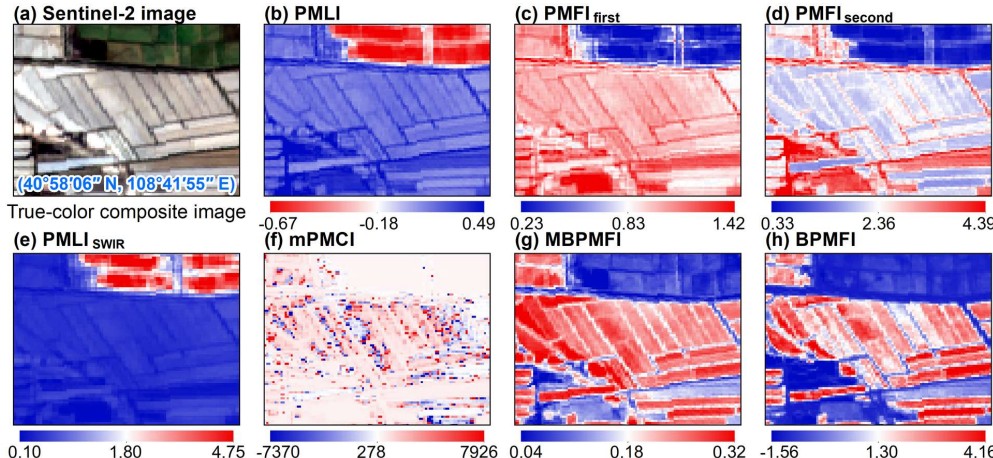


**Figure 8.**   Visual evaluation results based on the plastic-mulched farmland indices of PMLI (b), $PMFI_{first}$ (c), $PMFI_{second}$
(d), $PMLI_{SWIR}$ (e), mPMCI (f), MBPMFI (g), and BPMFI (h) in a representative area of Bayan Nur City, Inner Mongolia
Province (a). The red color is related to PMF, and the blue color is associated with backgrounds under all of the seven
indices.
**4.1.2 Visual comparison in Region II**
Unlike Region I, Region II is dominated by bare land and PMF, with PMF distributed sparsely in long
strips. MBPMFI and BPMFI continued to demonstrate the best performance in highlighting PMF and
suppressing background noise (Fig. 9 (g) and (h)). Since PMLI and $PMLI_{SWIR}$ were primarily proposed to
differentiate PMF from bare land (Lu et al., 2014; Hao et al., 2019), they could roughly identify PMF but
failed to suppress some backgrounds, such as roads and buildings (Fig. 9 (b) and (e)). In contrast to Region
I (Fig. 8), the reflectance values of PMF in $PMFI_{first}$ and $PMFI_{second}$ were both lower than those of Non-
PMF, indicating that Non-PMF was highlighted, while PMF was suppressed (Fig. 9 (c) and (d)). This was
because, compared with bare land, PMF had higher reflectance values in both the blue and near infrared
spectra (Xiong et al., 2019; Hasituya et al., 2016; Hao et al., 2019), but no obvious difference was observed
in the shortwave-infrared spectrum (Hasituya et al., 2016). Consequently, PMF might have lower values in
PMFI$_{first}$ and PMFI$_{second}$ than bare land. The mPMCI continued to exhibit the poorest performance among
the seven indices (Fig. 9 (f)). Compared with MBPMFI, BPMFI was more effective in suppressing Non-
PMF land covers, such as bare land and buildings (Fig. 9 (g) and (h)). In general, BPMFI still performed
the best in Region II.

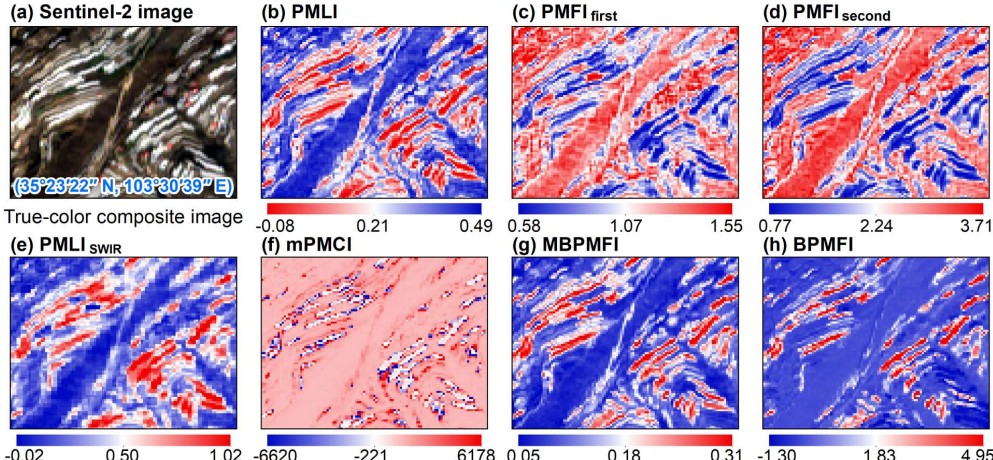

**Figure 9.** Visual evaluation results based on the plastic-mulched farmland indices of PMLI (b), PMFI$_{first}$ (c), PMFI$_{second}$
(d), PMLI$_{SWIR}$ (e), mPMCI (f), MBPMFI (g), and BPMFI (h) in a representative area of Linxia City, Gansu Province (a).
The red color is related to PMF, and the blue color is associated with backgrounds under all of the seven indices.
**4.1.3 Visual comparison in Region III**
Differing from Region I and Region II, Region III featured a more complex background environment,
including PMF, vegetation, bare land, buildings, and plastic greenhouses. Among the seven indices, BPMFI
demonstrated the best performance in identifying PMF and suppressing background noise (Fig. 10 (h)).
While PMLI, PMFI$_{first}$, and MBPMFI were also able to highlight PMF, they failed to effectively suppress
certain background elements, particularly buildings and plastic greenhouses (Fig. 10 (b), (c), and (g)).
PMFI$_{second}$ and PMLI$_{SWIR}$ seemed to wrongly highlight backgrounds while suppressing PMF (Fig. 10 (d)
and (e)). The mPMCI continued to perform the worst in recognizing PMF (Fig. 10 (f)). Overall, BPMFI
effectively separated PMF from background elements, whereas PMF detected by other indices was easily
confused with the background.
The evaluation results indicated that BPMFI had the highest robustness in identifying PMF under
various environmental conditions. It could more effectively highlight PMF and suppress Non-PMF than the
other six indices. The performance of MBPMFI ranked only next to BPMFI. Although MBPMFI was useful
for identifying PMF in Region I and Region II, it failed to suppress the complex backgrounds in Region III.
The other five indices were only effective in identifying PMF in specific regions, lacking the capability to
consistently recognize PMF under complicated background conditions.

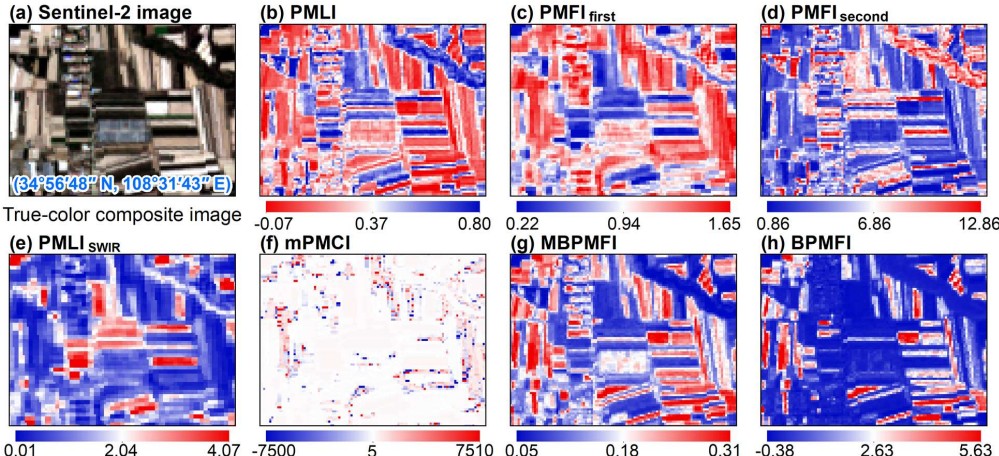


**Figure 10.** Visual evaluation results based on the plastic-mulched farmland indices of PMLI (b), $PMFI_{first}$ (c), $PMFI_{second}$
(d), $PMLI_{SWIR}$ (e), mPMCI (f), MBPMFI (g), and BPMFI (h) in a representative area of Xianyang City, Shaanxi Province
(a). The red color is related to PMF, and the blue color is associated with backgrounds under all of the seven indices.
**4.2 Classifier performance for the transferability scenarios**
All locally adaptive classifiers in the reference scenario (Scenario–Ref) showed satisfactory accuracy
in PMF mapping (Fig. 11 (a)). In this scenario, the F1 values of PMF ranged from 0.64 to 0.94, with an
average of 0.85. For all cities, the F1 values were close to or above 0.80, except in the NID (0.75), Baotou
(0.75), Hohhot (0.64), and Ordos (0.73). The high F1 values indicated that the locally adaptive classifiers
trained in this study can meet the practical demands for actual PMF mapping.
The F1 values from each transferability scenario (Scenario–T, Scenario–ST) were compared with
those from the reference scenario (Scenario-Ref) in each city, and the percentage change in the F1 values
($F1_{change}$) was also computed (Fig. 11 (b) and (c)). Generally, the comparison results showed that for all
cities, the F1 values decreased from the Scenario–Ref to Scenario–ST in an orderly manner (Scenario–ST >
Scenario–T > Scenario–Ref). In Scenario–T (Fig. 11 (b)), the average $F1_{change}$ was -2.92% for 2019 and -
6.07% for 2021. Except in Qinghai, Datong, and Yangquan, where the $F1_{change}$ values were below -20% in
2021, the $F1_{change}$ values for all other cities were above -20%. In Scenario–ST (Fig. 11 (c)), the average
$F1_{change}$ was -13.86% for 2019 and -21.94% for 2021, which was absolutely 10.94% and 15.87% lower than
in Scenario–T for 2019 and 2021, respectively. Furthermore, one-third of the cities in the Loess Plateau
showed a percentage decline in the F1 values over -20% in Scenario–ST. The greater decrease in F1 values



in Scenario–ST compared to Scenario–T indicated that Scenario–ST was hardly suitable for the mapping
of PMF distributions across different years.

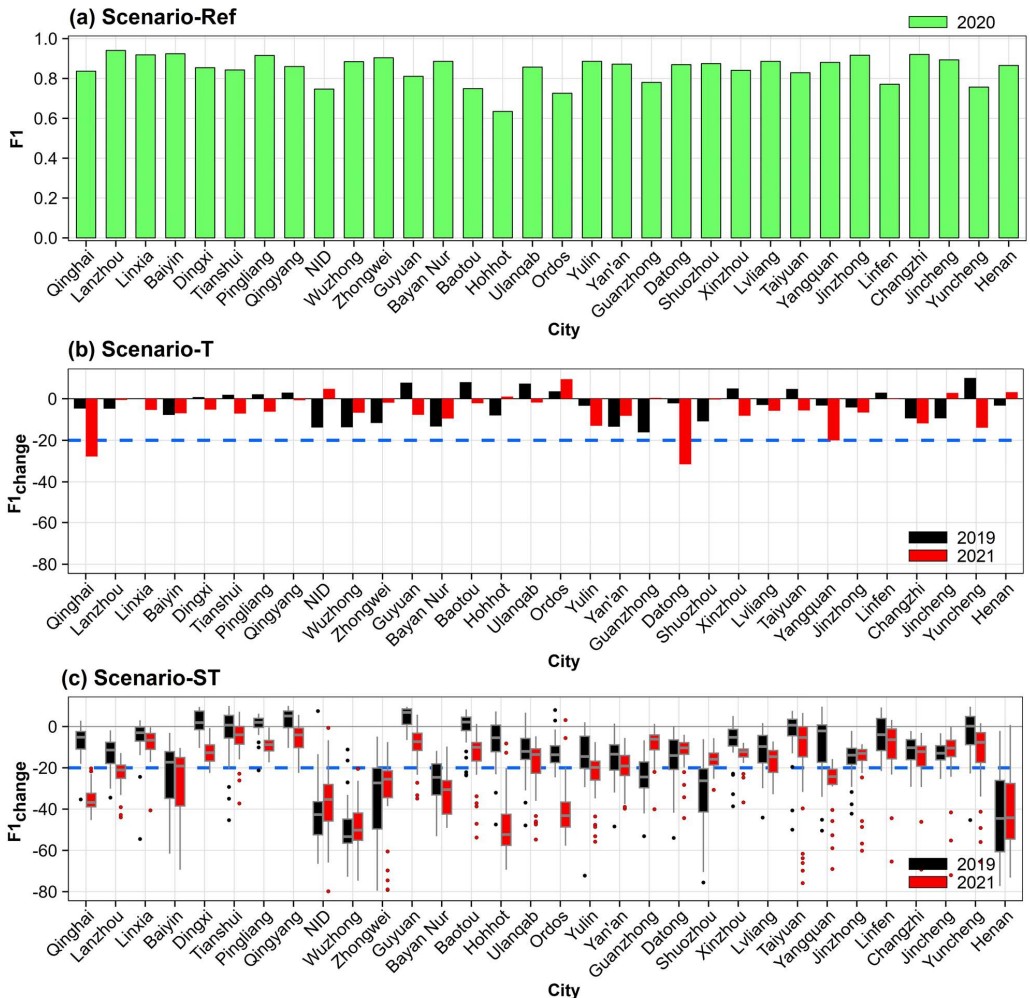


**Figure 11.**    Transferability evaluation of the pre-trained classifiers for plastic-mulched farmland (PMF) mapping. (a) F1
values of PMF mapping based on locally adaptive classifiers trained with data in 2020 in each city of the Loess Plateau
(Scenario–Ref). (b) Percentage change of F1 values ($F1_{change}$) for temporal transferability scenario (Scenario–T). (c)
$F1_{change}$ for spatial–temporal transferability scenario (Scenario–ST).

### 4.3 Accuracy assessment for resultant maps

Given the strong performance of the temporal classifier transfer (Scenario–T) in multi-year PMF
mapping, classifiers trained on automatically generated samples and optimal features in 2020 were
exploited to mapping PMF distributions (PMF-LP) for 2019–2021. The accuracy assessment for the three
years was implemented based on the validation samples of the entire study area (Table 3). The OA of the
resultant maps for all three years varied from 0.82 (2021) to 0.87 (2020). PMF was accurately identified,


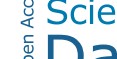

with F1 values ranging from 0.80 in 2021 to 0.86 in 2020, averaging 0.83 over the three years. For all three-
year classification results, PMF exhibited higher PA than UA, indicating that commission errors in PMF
recognition were higher than the omission errors. The higher commission errors indicated that some Non-
PMF pixels were incorrectly classified as PMF, potentially leading to an overestimation of PMF areas.
Detailed pixel-wise accuracy assessments for each city in the Loess Plateau were presented in Table S4~S6.
**Table 3.**
Confusion matrix of the plastic-mulched farmland maps of the Loess Plateau (PMF-LP) for the year 2019–2021. Reference
categories are rows while map categories are columns. The numbers in the confusion matrix are the number of ground
reference samples. The producer accuracy (PA), user accuracy (UA), F1-score (F1), and overall accuracy (OA) were also
displayed in the table. PMF: Plastic-mulched Farmland, Non-PMF: Non-plastic-mulched Farmland.

| Year | | PMF | Non-PMF | PA | UA | F1 | OA |
|------|---------|------|---------|------|------|------|------|
| 2019 | PMF | 2685 | 414 | 0.87 | 0.79 | 0.83 | 0.84 |
| | Non-PMF | 714 | 3278 | 0.82 | 0.89 | 0.85 | |
| 2020 | PMF | 4678 | 664 | 0.88 | 0.84 | 0.86 | 0.87 |
| | Non-PMF | 918 | 5880 | 0.87 | 0.90 | 0.88 | |
| 2021 | PMF | 2468 | 455 | 0.84 | 0.76 | 0.80 | 0.82 |
| | Non-PMF | 768 | 3023 | 0.80 | 0.87 | 0.83 | |


The estimated areas derived from our maps were compared with the statistical data from yearbooks of
12 cities for 2019–2021 (Fig. 12). The PMF areas derived from our maps exhibited high consistency with
the statistics, with coefficients of determination ($R^2$) of 0.92, 0.93, and 0.87 for the year 2019, 2020, and
2021, respectively. Notably, the resultant maps for 2021 (Fig. 12 (c)) tended to overestimate PMF areas
compared to the statistics, particularly in Qinghai and northern Shanxi, likely due to snow and cloud
residuals being misclassified as PMF (discussed in Section 5.3). The strong consistency between the
estimated areas and the statistical data underscored the reliability of the maps produced in this study.



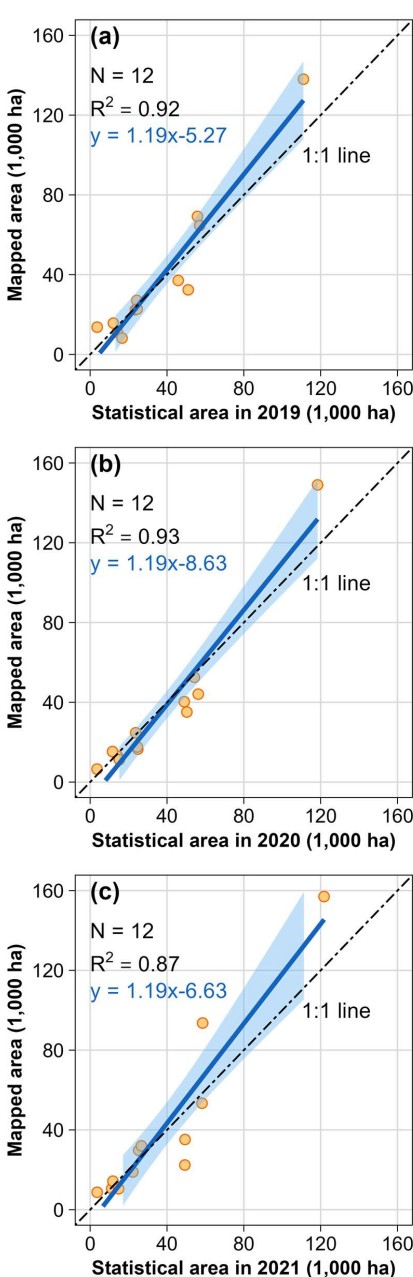

**Figure 12.** Comparison of the mapped PMF (Plastic-mulched Farmland) areas with the statistics for 12 cities with PMF area statistics for the year 2019 (a), 2020 (b), and 2021 (c).

## 4.4 Spatial distributions of PMF

The final PMF mapping results were shown in Fig. 13. To quantify the intensity of plastic film usage across different cities in the Loess Plateau, we calculated the ratio of plastic-mulched area (derived from





the resultant maps) to the total cultivated land area (derived from cropland layers in Section 2.2.2) for each
city (Fig. 13 (c)). PMF was widely distributed across the Loess Plateau, exhibiting a pattern of extensive
dispersion and localized clustering (Fig. 13 (a)). The Hetao Irrigation District, including the Bayan Nur and
Baotou cities in Inner Mongolia Province, exhibited the highest intensity of plastic film usage at 18%,
followed by Northern Shanxi (including Datong, Shuozhou, and Xinzhou) at 17%, and the Eastern Gansu-
Southern Ningxia region (including Dingxi, Baiyin, Pingliang, Qingyang, Zhongwei, and Guyuan) at 16%
(Fig. 13 (c)). In these arid and cold regions, plastic films have been extensively used for decades to ensure
crop yields by regulating temperature and conserving moisture (Wang et al., 2022a; Yan et al., 2014; Zhao
et al., 2023). The total PMF areas for each 0.10° longitude and latitude bin were also presented on the top
and right sides of the 2020 resultant map (Fig. 13 (a)). Plastic film was extensively used in latitude ranges
of 35°–37° N and 40°–41° N, and in the longitude ranges of 106°–109° E and 112°–114° E. However,
plastic film usage in the latitude range of 38°–40° N and longitude range of 109°–111° E was comparatively
lower than other regions. This disparity could be attributed to the prevalent local land cover types in Ordos
of Inner Mongolia Province, where grassland and bare land account for about 90% of the total area (Zhang
et al., 2021c; Yang and Huang, 2021).

To further validate the PMF mapping results, a more detailed visual evaluation was conducted by

selecting representative areas in different years (Fig. 14). Obviously, PMF was effectively distinguished
from other land cover types, regardless of various cropland sizes and shapes. In site A, Bayan Nur, Inner
Mongolia Province, PMF was densely distributed in regular rectangles (Fig. 14 (a)). There were clear
separations between PMF and roads in the mapping results. When the classifiers trained in 2020 were
applied to 2019 and 2021, some PMF pixels in site A were not identified in 2019 (Fig. 14 (b)), while in
2021, there were instances of erroneously classifying Non-PMF pixels as PMF (Fig. 14 (f)). Site B in
Xinzhou of Shanxi Province, had large fields with irregular shapes compared to site A (Fig. 14 (g))
compared with site A. The PMF mapping results from 2019 to 2021 at this site showed strong spatial
agreement with the Sentinel-2 true-color composite images (Fig. 14 (h), (j), and (l)). In site C, Xianyang,
Shaanxi Province, PMF was scattered among winter crops (Fig. 14 (m)). The mapping results showed good
separability between winter crops and PMF across different years (Fig. 14 (n), (p), and (r)). For site D in
Linxia of Gansu Province, the cultivated farmland was primarily characterized by terraced fields, presenting
elongated and irregular shapes (Fig. 14 (s)). These terraced fields were interspersed with vegetations,
buildings, and bare lands. The mapping results were also satisfactory in this site (Fig. 14 (t), (v), and (x)).

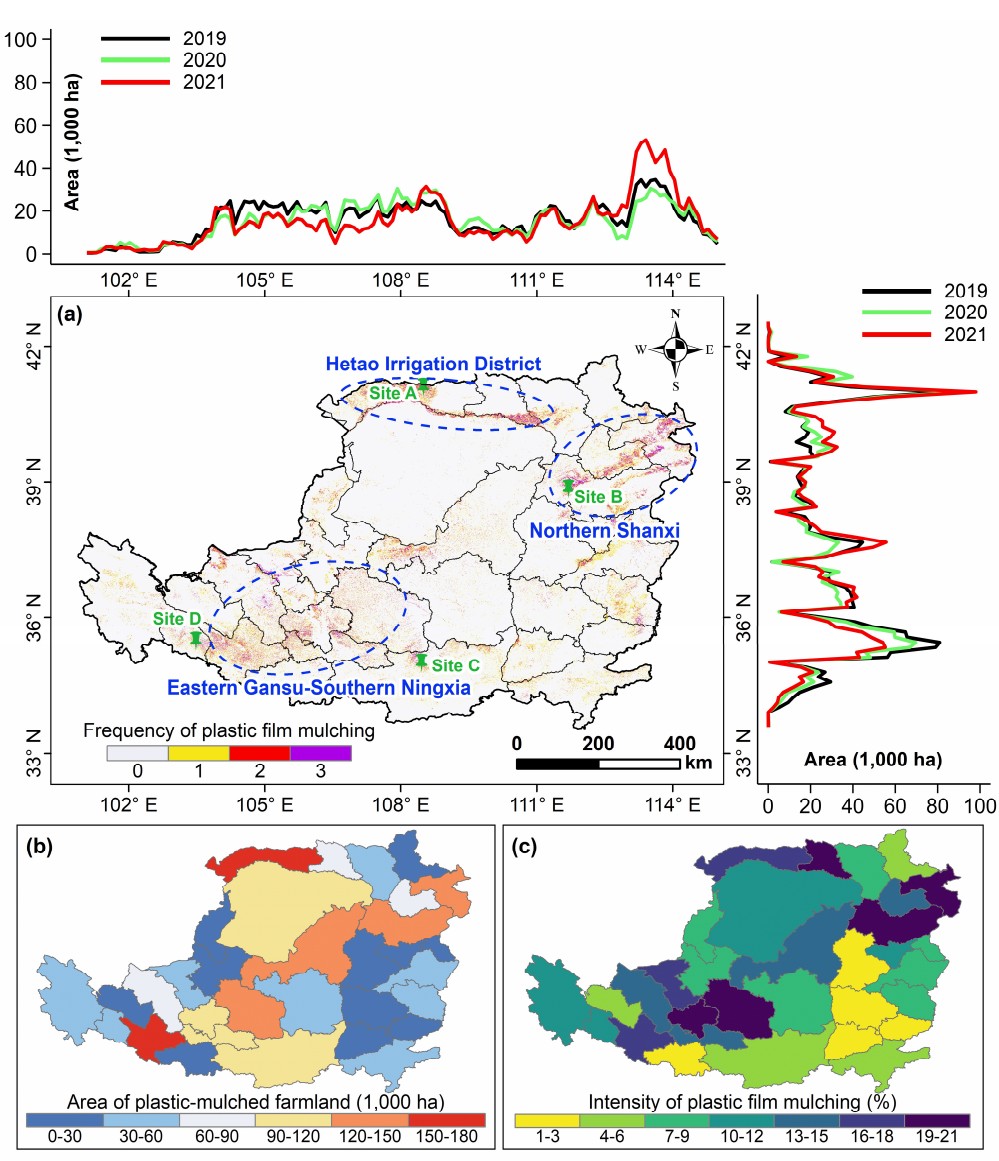

**Figure 13.**    Spatial patterns of plastic-mulched farmland (PMF) in the Loess Plateau from 2019 to 2021. (a) Frequency of plastic film mulching. The blue dashed ellipses represent the regions with dense distributions of PMF. Site A, B, C, and D are the zoom-in view cases in Fig. 14. (b) Area of PMF in each city in 2020. (c) Intensity of plastic film mulching in 2020 across various cities. Intensity of plastic film mulching is defined as the ratio of PMF area to total cultivated area.



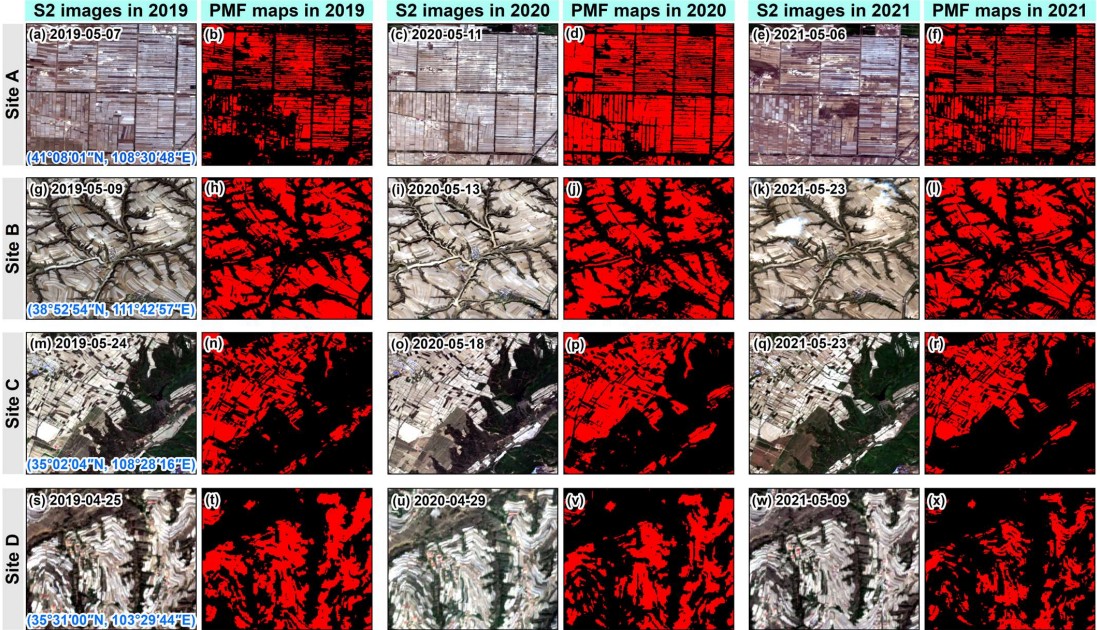

**Figure 14.**   Spatial details of the plastic-mulched farmland (PMF) map subsets from different regions in the years of 2019, 2020, and 2021. Site A, B, C, and D correspond to the four typical regions marked in Fig 13. The 1st, 3rd, and 5th columns are the Sentinel-2 (S2) images with a true-color composite of red, green, blue bands in the years of 2019, 2020, and 2021, respectively. The 2nd, 4th, and 6th columns are the PMF maps established in this study.

## 5  Discussion

### 5.1 Feasibility of MBPMFI and BPMFI for automatic training sample generation

Image sources, algorithms, and training samples are three crucial factors influencing any land cover classification task (Huang et al., 2020; Wen et al., 2022; Zhang et al., 2021b). The advancement of remote sensing technology has provided access to massive open datasets and powerful cloud-based computing platforms (Song et al., 2017; Yang et al., 2019; Tamiminia et al., 2020). However, large-scale and multi-year land cover classification is still constrained by the lack of training samples (Huang et al., 2020; Wen et al., 2022). This issue is particularly pronounced for land cover types like PMF, which have received less attention in the past, making training sample collection more challenging. To address this gap, we developed two novel indices, MBPMFI and BPMFI, to automatically generate training samples. Compared with traditional field surveys and visual interpretation, the proposed indices offer a more convenient and labor-saving alternative for obtaining training samples. The prior information, such as mulching and flourishing dates, is not strictly time-specific and can be easily obtained from local management experiences and relevant studies. Furthermore, because the two indices rely solely on the blue band, their calculation is simple and fast.





Additionally, the visual comparison results under various environmental conditions in this study
demonstrated the superior robustness of MBPMFI and BPMFI over the other existing indices in
highlighting PMF signals and suppressing background noises. Previous studies have shown that PMF can
be effectively distinguished from other land cover types across the visible to shortwave-infrared spectrum
during the mulching stage (Lu et al., 2014; Hasituya et al., 2016; Hao et al., 2019). However, PMF indices
based solely on optical images from the mulching stage often struggle to differentiate PMF from bare land
and impervious surfaces (Lu et al., 2014; Hao et al., 2019) (Fig. 6 (e)~(h)), which limits their applicability
in complex environments. In this study, we considered PMF characteristics across the pre-mulching,
mulching, and flourishing stages to develop two novel PMF indices. Both indices exhibited greater
robustness than the existing ones in distinguishing PMF from various complex backgrounds. Of the two
indices, BPMFI was more effective in highlighting PMF signals and suppressing background noise, making
it particularly promising for rapid and precise PMF recognition in areas with seamless satellite imagery.
However, because the calculation of BPMFI requires images from three different periods, which might be
unavailable in cloudy regions, MBPMFI is more practical for PMF recognition when only mulching-stage
images are available. To further improve the quality of training samples in this study, we collected samples
classified as PMF by both indices, and then conducted strict spatial filtering. The high PMF mapping
accuracy (F1 = 0.86) and the strong correlation ($R^2$ = 0.93) between estimated and statistical PMF areas in
2020 confirm the quality of these training samples.
**5.2 Threshold stability of the MBPMFI and BPMFI**
Threshold used for index-based classification is typically determined using statistical data (Zhang et
al., 2022e; Zhang et al., 2022b), ground truth data (Zhang et al., 2022g; Zhou et al., 2024), or automatic
threshold determination methods (Yang et al., 2023). In this study, we employed ground truth samples from
rectangular regions (Fig. 1 (a)) to determine the threshold for each province of the Loess Plateau, because
statistical data were not available in most cities and automatic methods could require significant
computational resources at large scales. To evaluate the stability of MBPMF and BPMFI thresholds, we
tested the mapping accuracy in the rectangular regions based on thresholds -50%–50% off the best threshold
(Fig. 15). Compared with MBPMFI, BPMFI can achieve more stable performance even with a threshold
largely off from the best threshold. The F1 values ranged from 0 to 0.98 for MBPMFI and from 0.80 to
0.97 for BPMFI with the thresholds ±50% off from the best threshold across all regions. The stable F1
values of BPMFI indicated that it is not sensitive to threshold variations, which makes it well-suited for
large-scale PMF mapping.
According to the results of our study, the best BPMFI threshold was 0.40 for Qinghai, Shaanxi, and
Shanxi, and 0.80–1.10 for Gansu, Ningxia, and Inner Mongolia. The relatively large difference in the best
thresholds may be attributed to variations in plastic materials (Xiong et al., 2019; Zhang et al., 2022g) and
crop species grown in the fields. The former affects the peak reflectance of PMF during the mulching stage,
and the latter influences the absorption of the visible spectrum during the flourishing stage. The more
detailed relationship between these factors and the spectral signatures of PMF deserves to be explored in
the future. Additionally, based on the threshold stability assessment in Fig. 15, we recommend using a
BPMFI threshold of 0.40–0.60 (i.e., approximately 50%–150% of the optimal threshold) and an MBPMFI
threshold of 0.14–0.15 (i.e., approximately 90%–110% of the optimal threshold) for PMF recognition
across the entire Loess Plateau. But for high-precision PMF mapping, it is suggested to segment the whole
area into sub-regions and determine the optimal thresholds separately for each region.

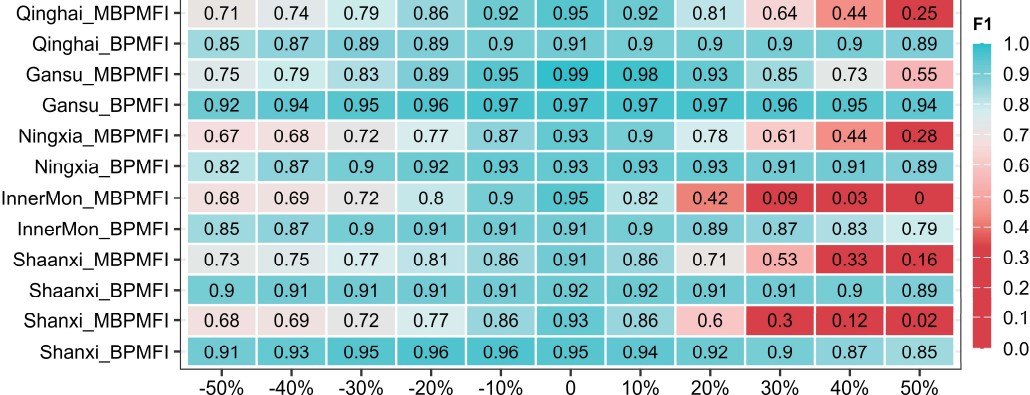


**Figure 15.** Plastic-mulched farmland (PMF) mapping accuracies (F1-score) based on thresholds -50%–50% off the best
threshold in the two novel PMF indices of MBPMFI and BPMFI. "±n%" indicates threshold = best threshold ±n% × best
threshold.
**5.3 Feasibility and error of classifier transfer for PMF mapping**
Classifiers, pre-trained in specific areas/years with sufficient samples, have great potential for rapid
land cover and crop type mapping in new areas/years where training data are lacking (Ma et al., 2024).
Previous studies on classifier transfer primarily focused on crop type mapping (Wang et al., 2019; Hao et
al., 2020; Yang et al., 2023). In contrast, ground reference samples for PMF are more limited, and the
feasibility of employing classifier transfer for PMF mapping has not been thoroughly explored.
Understanding the extent of performance loss in classifier transfer is crucial for large-scale and multi-year
PMF mapping. Therefore, this study quantitatively evaluated the performance loess when the trained
classifier is directly applied in two transferability scenarios (Scenario–T, Scenario–ST) for PMF mapping.
The findings demonstrated decreased performance in both transferability scenarios compared to the no-
transfer case (Scenario–Ref). The temporal classifier transfer (Scenario–T) was more suitable for retracing





historical PMF distributions than spatial–temporal classifier transfer (Scenario–ST), with percentage
changes in F1 being less than 7.0% on average compared to the no-transfer case. Several potential factors
that may cause performance reduction in Scenario–T and Scenario–ST (Fig. 11) could be categorized into
three main groups: (1) missing clear Sentinel-2 data during the mulching stage; (2) snow and cloud residual
contaminations during the pre-mulching stage; and (3) variations in mulching dates across different regions
and years.

First, high-quality Sentinel-2 data during the mulching stage is crucial for enlarging the inter-class

variations between PMF and Non-PMF as illustrated in Section 3.1.2. Particularly, the local peak values in
the blue band of time-series Sentinel-2 (i.e., Blue_peak) is the most important feature for PMF mapping
(Fig. 7). Missing data during the mulching stage could result in the disappearance or distortion of the local
peak values, causing classifiers trained on regions/years with high-quality data to fail in mapping PMF in
regions/years with missing data. For example, during the mulching stage, more than half of the pixels in
Qingyang had more than four clear-sky observations (Fig. 16 (a)), while nearly 75% of the pixels in Henan
had fewer than three clear-sky observations (Fig. 16 (b)). Consequently, the local peaks of the harmonic
regression curves in Henan (Fig. 16 (e)) were lower than those in Qingyang (Fig. 16 (d)) during 2019–2021,
which led to low accuracy when using classifiers trained with Qingyang data from 2020 to recognize PMF
in Henan for 2019 (F1 = 0.64) and 2021 (F1 = 0.49). Although the Sentinel-2 satellite provides data with a
5-day revisit cycle and cloud contamination in northern China is less severe than in southern China, the
observable duration of plastic film is short and influenced by crop canopy, which leads to the unavailability
of images containing plastic film signals. Multi-sensor satellite data could be employed to increase the
density of observations to account for this limitation.

Next, clouds and snow, which exhibit similar bright characteristics in all visible bands as PMF, are

difficult to be thoroughly masked out using only the quality assessment (QA) bands of Sentinel-2 (You and
Dong, 2020). For example, although the percentage of clear-sky observations in Qinghai was the same as
in Qingyang (Fig. 16 (a) and (c)), snow and cloud residuals distorted the harmonic regression curves in
Qinghai in 2021 (Fig. 16 (f) and (g)), leading to the performance loss when using classifiers trained with
Qinghai (or other city) data from 2020 to recognize PMF in the same region for 2021 (Fig. 11 (b) and (c)).
To mitigate the negative effects of clouds and snow on classifier transfer for PMF mapping, more effective
cloud detection algorithms should be implemented. Inspired by You and Dong (2020), two spectral indices
(Normalized Difference Moisture Index (NDMI) and Normalized Difference Snow Index (NDSI)) could
be used to mask out clouds to improve the performance of classifier transfer in the future.

Finally, variations in sowing dates of plastic-mulched crops among different regions could lead to

changes in mulching dates, which in turn affect the occurrence dates of local peaks and the shape of
harmonic regression curves. In this study, in the Ningxia Irrigation District (NID, Fig. S1) and Wuzhong of



Ningxia Province, mulching dates were earlier than in other cities (Fig. S11) due to the widespread
cultivation of watermelon, which is mainly sown in March. Consequently, the performance exhibited a
significant decrease (median $F1_{change} \approx$ -50%) when using classifiers trained with data from other city in
2020 to recognized PMF in the two cities (Scenario–ST) for 2019 and 2021.

Apart from the main factors mentioned above, climate variables (e.g., rainfall, temperature), farming

practices (e.g., sowing and harvesting practices), technological evolution, and policy implementation could
also impact the mulching practices, which may adversely affect the performance of classifier transfer. The
links between classifier transferability and these factors require further exploration. Additionally, multi-
year/multi-region data have been employed to enhance classifier transferability for crop type mapping
(Wijesingha et al., 2024; Orynbaikyzy et al., 2022; Luo et al., 2022). Considering the increasing focus on
PMF mapping (Veettil et al., 2023), there will likely be more training data from different years and regions
available to develop more generalized classifiers for PMF mapping in the future.

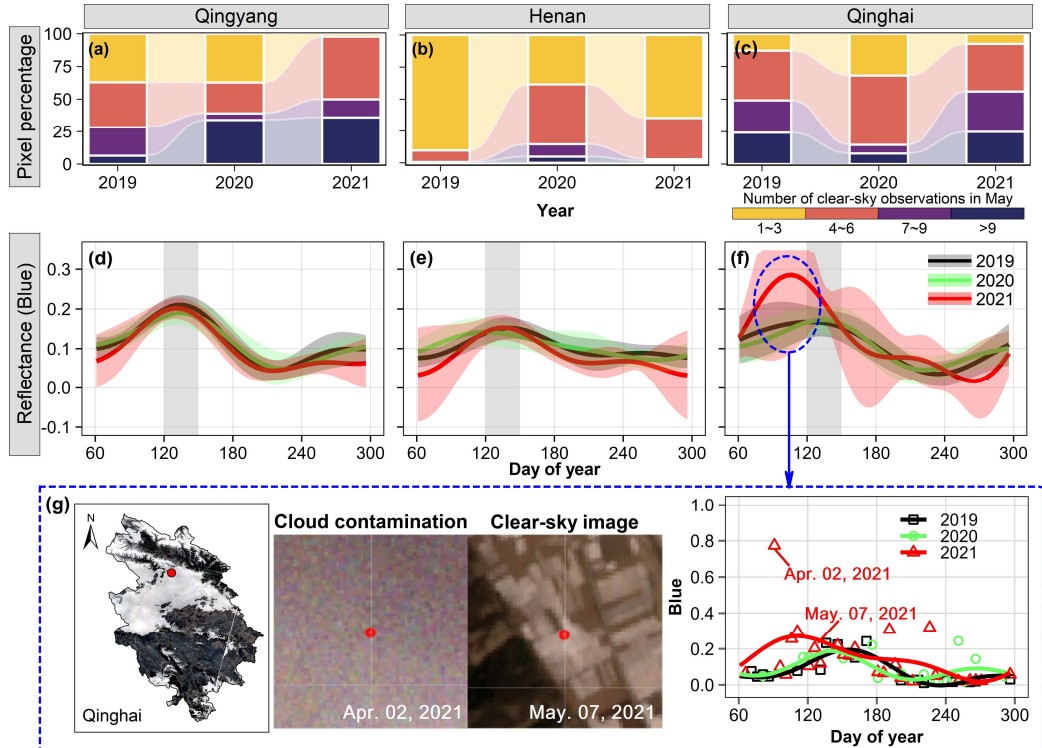


**Figure 16.**   Explanations for the failures of Plastic-mulched farmland (PMF) mapping caused by data missing and cloud
contamination. (a)~(c) Percentage of clear-sky observations at each cropland pixel during the mulching stage in Qingyang,
Henan, and Qinghai. (d)~(f) Harmonic regression curves of PMF in Qingyang, Henan, and Qinghai for the year 2019,
2020 and 2021. The gray buffer represents mulching period. The black, green, and red buffers indicate one standard
deviation. (g) A representative example in Qinghai for depicting the adverse effects of snow and clouds on PMF mapping.



### 5.4 Advantages and uncertainties of the new framework

In this study, we proposed a new framework that combines index-based methods with supervised classifier transfer for large-scale PMF mapping. This framework offers two main advantages. First, we introduced two novel and robust PMF indices (MBPMFI and BPMFI) for generating training samples, significantly enhancing the efficiency of sample collection compared to field surveys and visual interpretation. Second, temporally transfer pre-trained classifiers to mapping multi-year PMF distributions further facilitates the long-term PMF monitoring without training domain- and year-specific classifiers. The automatic generation of training samples and classifier transfer make this framework highly promising for rapidly mapping large-scale and multi-year PMF distributions in practice. The satisfactory mapping accuracy of PMF (F1 > 0.80) in the Loess Plateau during 2019–2021 further demonstrated the reliability of this framework. Additionally, since the reflectance bands used for training sample generation and feature extraction are similar to those of Landsat satellites, this framework is also believed to be applicable for PMF mapping with Landsat data.

Although the new framework successfully identified PMF across different regions and years in the Loess Plateau of China, there still exists some room for improvement. First, two different cropland layers from 2020 were integrated to exclude the non-cropland pixels for 2019–2021 before PMF recognition. However, errors in these products may affect PMF mapping results. For instance, unrecognized cropland pixels in the products could lead to the loss of potential PMF pixels, thereby increasing the omission errors in the PMF mapping results. One possible solution is to integrate more state-of-the-art land cover products (Tu et al., 2024; Zhang et al., 2024) to enhance the reliability of the cropland mask.

Next, although Sentinel-2 has a 5-day revisit cycle, clear-sky observations are still often limited in some regions, especially in southern China, due to inevitable cloud contaminations. The large-scale loss of images during the mulching stage could constrain the application of the proposed framework. To address this issue, more optical satellite imagery, such as Landsat-4/5/7/8/9, could be integrated to provide denser time-series observations in the future. Furthermore, this study did not include radar data which is impervious to atmospheric interference in PMF mapping. Incorporating radar data for PMF mapping will be the next step of our research.

Third, this study focused on evaluating temporal and spatial-temporal transferability of the RF classifier for PMF mapping. Due to the limitations of GEE platform, deep learning algorithms such as Deep Neural Network (DNN) and Convolution Neural Network (CNN), which exhibit better performance than the traditional supervised classifiers (Zhong et al., 2019; Wang et al., 2021), were not explored in this study. However, employing deep learning algorithms for PMF mapping presents challenges related to





computational costs and the need for more training data (Xu et al., 2021; Zhang et al., 2020a; Wijesingha
et al., 2024).

**6  Data availability**

The PMF-LP product generated in this paper is openly available at
https://doi.org/10.5281/zenodo.13369426 (Zhao et al., 2024). The dataset includes a set of GeoTIFF images
in the EPSG:4326 spatial reference system. The value 0 and 1 represent PMF and Non-PMF, respectively.
We encourage users to independently verify the PMF maps.

**7  Conclusions**

Plastic film mulching technology can effectively enhance crop yield, but also poses significant
environmental issues, such as "white pollution" and soil microplastic contamination. Unfortunately, there
is no method currently available for the rapid and automated mapping of plastic-mulched farmland (PMF)
across large areas and multiple years. In this study, to address the challenge of lacking ground reference
samples to train supervised classifiers, we developed an automatic training sample generation method based
on two newly proposed PMF indices: MBPMFI and BPMFI. To rapidly retrace historical PMF distributions,
we thoroughly explored the spatial–temporal transferability of pre-trained classifiers and the factors
influencing their transferability. Finally, PMF distribution maps for the Loess Plateau of China were
produced by integrating the automatic training sample generation method with the classifier transfer
approach.
The results of this study indicated that the novel PMF indices, MBPMFI and BPMFI, outperformed
the existing PMF indices in PMF recognition. The novel indices could not only highlight PMF information
but also suppress complex background signals under various environmental conditions. Temporal classifier
transfer (Scenario–T) showed great potential for rapid mapping of multi-year PMF distributions in regions
and years lacking ground reference samples, with an average performance loss less than 7.0% compared to
the locally adaptive classifiers. Based on the index-based training samples generation method and classifier
transfer, PMF distributions were generated across the Loess Plateau (PMF-LP) and achieved satisfactory
accuracy, with F1 values of 0.83, 0.86, and 0.80 for 2019, 2020, and 2021, respectively. The estimated PMF
areas also agreed well with the municipal-level agricultural census data ($R^2 \geq 0.87$). In summary, the newly
proposed PMF mapping framework has great potential to rapidly mapping PMF distributions across large
areas and multiple years. Future work will apply this framework to monitor the spatiotemporal dynamics
of PMF distributions in China.



**Supplement.**


The supplement related to this article is available online at: https://doi.org/

**Author contributions.**


CZ and JH conceptualized the study. CZ designed and carried out the experiment. CZ wrote the
original manuscript. YL, XC, and LW collected dataset and implemented formal analysis. ZW, HF, and QY
provided direction and comments. JH was responsible for funding acquisition. All authors discussed the
results and revised the manuscript.

**Competing interests.**


The contact author has declared that none of the authors has any competing interests.

**Disclaimer.**


Publisher's note: Copernicus Publications remains neutral with regard to jurisdictional claims in
published maps and institutional affiliations.

**Acknowledgements.**


The author gratefully acknowledgements all data providers whose data have been used in this study.
Then we thank the support of the Google Earth Engine platform, which provided essential data and
computational resources for this study.

**Financial support**


This research was supported by National Natural Science Foundation of China (No. 52079115), the
National Key Research and Development Program of China (No. 2021YFD1900700), and the "111 Project"
(No. B12007) of China.

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
