# Peer review of "PMF-LP: the first 10 m plastic-mulched farmland distribution map (2019-2021) in"

_Earth System Science Data, 2024_

## Referee Comment (RC1)

**1. General Comments:**

This study tried to generate PMF samples through spectral indices and then obtain the PMF maps on the Loess Plateau of China for 2019 to 2021 using random forest. However, the data quality is unsatisfactory after careful evaluation while the study area is too small. The method to automatically generate the PMF samples has certain limitations while more samples from in-situ field surveys is a must to increase the classification performance and to justify the reliability of the PMF data product.

As for data quality, there exist large false positives as bare land and impervious surfaces like airport are misclassified into PMF while the salt-and-pepper effect is not tackled. The temporal consistency among the three years is also an issue. More specific comments and the data quality check details are in the PDF uploaded.

**2. Specific Comments**

**(1) Limitations of the PMF indices**

Line 305 - line 317, two PMF indices (i.e., MBPMFI and BPMFI) are defined to enhance PMF regions while suppress background noise, however, several limitations are as follows. Since these indices are constructed mainly based on the blue band, in complex and fragmented agricultural environments, relying only on the blue band is insufficient. For example, in regions near water bodies or regions with a high soil moisture, the blue band is prone to interference, leading to the misclassifications of background as PMFs, or vice versa, reducing the reliability of training samples.

**(2) Limitations of determining the optimal index threshold**

In this study, provincial-specific thresholds are used by collecting sample points in rectangular areas, however, the use of rectangular regions may introduce bias and may not represent the entire province. Even within the same province, factors such as topography and crop planting habits can lead to significant variations in the spectral curves of PMF. Therefore, the optimal thresholds determined in this study could result in large errors when applied to the entire province, greatly weakening the reliability of the training samples.

For instance, the limitations discussed above would be serious in Shaanxi Province. Supposing that a rectangular area is selected for sample collection in the Guanzhong Plain, where the terrain is rather flat while wheat is the main crop planted, farmlands are large and orderly, and irrigation conditions are good. The optimal threshold in this region could not be the optimal or best in northern Shaanxi. As is known, northern Shaanxi is dominated by loess hilly and gully terrain, with fragmented land, diverse crops such as millet and sorghum in addition to wheat, and irrigation is difficult due to the terrain. In this case, using the optimal threshold determined from Guanzhong Plain to identify PMF regions in northern Shaanxi could lead to misclassifications. This greatly weakens the reliability of the training samples and would lower the accuracy of the PMF maps generated across the entire Shaanxi Province.

**(3) Limitations of optimizing training samples**

Ln line 305 - line 317. The eight-neighbor filter used is too simple which could not yield high performance especially in complex agricultural landscapes where land cover types exhibit high complexity and variability. For instance, small PMF regions being surrounded by heterogeneous

pixels would be ruthlessly excluded by the eight-neighbor. Therefore, it is highly probable to miss samples, which could seriously impact the PMF mapping results.

Actually, the method of determining a high-quality sample when all the eight neighboring pixels belong to the same category has its flaws. For example, mixed pixels between PMF and bare land are common, where bare land is easily sampled as a high-confidence sample, leading to a large number of high-reflectance bare land pixels being incorrectly classified as PMF. Additionally, when using threshold methods, impervious surfaces such as white rooftops, could share similar spectral characteristics with PMF and would be misclassified as plastic mulch. More critically, the entire sample screening process does not make use of expert information, lacking of agricultural knowledge guidance and verification. This further reduces the reliability of the samples, making the PMF maps showing unsatisfactory details.

**(4) Limitations of statistical data**

In section 2.2.4, the estimated PMF area is compared with the statistical data from the agricultural department. However, the statistical bureaus mainly focused on the portion directly purchased and used in agricultural production, with a lack of refinement and comprehensiveness in the statistics on plastic mulch usage. Therefore, the statistical data on plastic mulch provided by statistical departments can only serve as a reference and cannot accurately represent the actual plastic mulch used in the fields. Such data typically overestimates the actual amount of plastic mulch used.

**(5) Unsatisfactory PMF maps**

The PMF maps provided by the authors are unsatisfactory, while existing obvious misclassifications of bare land, building rooftops, plastic greenhouses, and the salt-and-pepper effect is serious. For details, refer to Section 2.

**2. Validation of the provide PMF maps**

**(1) Bare land misclassified as PMF**

Large areas of bare land are misclassified as plastic film. Meanwhile, salt-and-pepper effect is serious through the following figures.

Site A. Coordinates: 35° 58' N, 109° 33'E. 2019 PMF map.

[Figure]

Fig. 1. (a) PMF in 2019. *Site A: 35° 58' N, 109° 33'E*

[Figure]

Fig. 1 (b) Sentinel-2 Satellite Imagery from April 15, 2019, to May 15, 2019.
*Site A: 35° 58' N, 109° 33'E.*

**(2) Impervious surfaces misclassified as PMF**

Impervious surfaces (e.g., airport) are misclassified as plastic film.

Site B. Coordinates: 34°49′N, 109°32′E. 2020 PMF map.

[Figure]

Fig. 2. (a) PMF in 2020. *Site B: 34°49'N, 109°32'E.*

[Figure]

Fig. 2. (b) Sentinel-2 Satellite Imagery from April 15, 2020, to May 15, 2020.
*Site B: 34°49'N, 109°32'E.*

Site B. Coordinates: 34°49′N, 109°32′E. 2021 PMF map.

[Figure]

Fig. 3. (a) PMF in 2021. *Site B: 34°49'N, 109°32'E.*

[Figure]

Fig. 3. (b) Sentinel-2 Satellite Imagery from April 15, 2021, to May 15, 2021.
*Site B: 34°49'N, 109°32'E.*

**(3) Temporal inconsistency**

For the same area, the PMF map in 2021 differs significantly from those in 2019 and 2020, which is questionable since the crop planting habits is not that changeable in the study area, as shown in the Fig. 4 to Fig. 6. The temporal consistency and the Sentinel-2 images both show that there are large bare land regions misclassified into PMF.

(1) Site C. Coordinates: 37°21′N 112°24′E. 2021 PMF map.

[Figure]

Fig. 4. (a) PMF in 2021. *Site C: 37°21'N 112°24'E.*

[Figure]

Fig. 4. (b) Sentinel-2 Satellite Imagery from April 15, 2021, to May 30, 2021.
*Site C: 37°21'N 112°24'E.*

(2) Site C. Coordinates: 37°21′N 112°24′E. 2020 PMF map.

[Figure]

Fig. 5. (a) Overlay PMF in 2020 and the Google MAP (very few PMF witnessed in 2020 when compared with 2021). *Site C: 37°21′N 112°24′E.*

[Figure]

April 15, 2020 - May 15, 2020          May 15, 2020 - May 30, 2020

Fig. 5. (b). Sentinel-2 Satellite Imagery in 2020 (Via GEE). *Site C: 37°21′N 112°24′E.*

(3) Site C. Coordinates: 37°21′N 112°24′E. 2019 PMF map.

[Figure]

Fig. 6. (a). Overlay map of the classification results of PMF in 2019 and the Google MAP (very few PMF witnessed in 2019 when compared with 2021). *Site C: 37°21′N 112°24′E.*

[Figure]

[Figure]

April 15, 2019 - May 15, 2019            May 15, 2019 - May 30, 2019

Fig. 6. (b). Sentinel-2 Satellite Imagery in 2019 (Via GEE). *Site C: 37°21'N 112°24'E.*

**(4) False positives are too high**

There is an issue of over-segmentation of PMF, where surrounding background is identified as plastic mulch. See Fig. 7 to Fig. 9 for details.

(1) Site D. Coordinates: 40°51′N, 107°45′E. 2019 PMF map.

[Figure]

Fig. 7. (a) Classification Results of PMF in 2019. *Site D: 40°51'N, 107°45'E.*

[Figure]

[Figure]

April 15, 2019 - May 15, 2019            May 15, 2019 - May 30, 2019

Fig. 7. (b) Sentinel-2 Satellite Imagery in 2019 (Via GEE). *Site D: 40°51'N, 107°45'E.*

(2) Site E. Coordinates: 40°39′N, 107°23′E. 2021 PMF map.

[Figure]

Fig. 8. (a) PMF in 2019. *Site E: 40°39'N, 107°23'E.*

[Figure]

April 15, 2020 - May 15, 2020          May 15, 2020 - May 30, 2020

Fig. 8. (b) Sentinel-2 Satellite Imagery in 2019 (Via GEE). *Site E: 40°39'N, 107°23'E.*

(3) Site F. Coordinates: 35°32′N, 105°36′E. 2021 PMF map.

[Figure]

Fig. 9. (a) PMF in 2019. *Site F: 35°32'N, 105°36'E.*

[Figure]

April 15, 2021 - May 15, 2021          May 15, 2021 - May 30, 2021

Fig. 9. (b) Sentinel-2 Satellite Imagery in 2021 (Via GEE). *Site F: 35°32′N, 105°36′E.*